# AI-enabled, implantable, multichannel wireless telemetry for photodynamic therapy

Woo Seok Kim [1,6], M. Ibrahim Khot [2,6], Hyun-Myung Woo[1,6], Sungcheol Hong [1], Dong-Hyun Baek[3], Thomas Maisey [2], Brandon Daniels[1], P. Louise Coletta[2], Byung-Jun Yoon [1,4✉], David G. Jayne [2✉] & Sung Il Park [1,5✉]

Photodynamic therapy (PDT) offers several advantages for treating cancers, but its efficacy is highly dependent on light delivery to activate a photosensitizer. Advances in wireless technologies enable remote delivery of light to tumors, but suffer from key limitations, including low levels of tissue penetration and photosensitizer activation. Here, we introduce DeepLabCut (DLC)-informed low-power wireless telemetry with an integrated thermal/light simulation platform that overcomes the above constraints. The simulator produces an optimized combination of wavelengths and light sources, and DLC-assisted wireless telemetry uses the parameters from the simulator to enable adequate illumination of tumors through high-throughput (<20 mice) and multi-wavelength operation. Together, they establish a range of guidelines for effective PDT regimen design. In vivo Hypericin and Foscan mediated PDT, using cancer xenograft models, demonstrates substantial suppression of tumor growth, warranting further investigation in research and/or clinical settings.

[1] Department of Electrical and Computer Engineering, Texas A&M University, College Station, TX, USA. [2] Leeds Institute of Medical Research, University of Leeds, Leeds, UK. [3] Department of Display and Semiconductor Engineering, Sun Moon University, Asan-si, Republic of Korea. [4] Computational Science Initiative, Brookhaven National Laboratory, Upton, NY, USA. [5] Institute for Neuroscience, Texas A&M University, College Station, TX, USA. [6] These authors contributed equally: Woo Seok Kim, M. Ibrahim Khot, Hyun-Myung Woo. ✉email: bjyoon@ece.tamu.edu; D.G.Jayne@leeds.ac.uk; sipark@tamu.edu

Photodynamic therapy (PDT) is an anti-cancer treatment, which has the advantage of targeting cancer cells whilst minimizing toxicity to normal healthy tissue[1,2]. PDT involves the administration of a photosensitizer, which is preferentially taken up and retained in tumors. Activation of the photosensitizer is mediated by light of a specific wavelength. In the clinical setting, this is conventionally achieved through the use of high-powered lasers. In the presence of oxygen, the excited photosensitizer can lead to the production of reactive oxygen species, inducing intracellular oxidative stress and triggering tumor cell death. Oxygen, a photosensitizer, and light are needed for PDT, and their relative importance is supported by in vivo and in vitro studies into the mechanisms of phototoxic cell death[3–5]. For example, tissue oxygenation in tumors is critical for the production of singlet oxygen and successful phototoxicity, with tumor hypoxia being a limiting factor to PDT efficacy PDT[6,7]. Similarly, the delivery and selective uptake of photosensitizer into tumors is vital for optimal PDT efficacy, whilst reducing toxicity to normal cells[8,9]. Finally, light delivery, and the use of fractionated light to minimize oxygen depletion, is important when designing effective photodynamic regimens[10].

Existing hardware PDT, in particular the light source, suffers from key limitations in their ability to: (1) deliver light to deep tumor cells (e.g., physical tethers to light sources would complicate access to targeted tumor cell), (2) control light sources and/or wavelengths in a programmed manner (e.g., light sources with single wavelength would activate a photosensitizer at a modest rate if a photosensitizer has two absorption peaks), and (3) be multiplexed so that multiple animals can be studied in parallel.

Recent advances in wireless technologies have enabled wireless control of light delivery directly to tumor cells in a freely behaving animal[11–13]. Biocompatible, miniaturized optoelectronic devices utilize electromagnetic (EM) wave propagation at a high frequency (HF) or ultra-high frequency (UHF) range to convert radio frequency (RF) energy into optical energy using a single wavelength of light sources[14–16]. Although these approaches have benefits, the activation of a photosensitizer by a single wavelength of light sources is a limitation. For example, Foscan has two peaks at 406 nm and 652 nm in absorption spectra according to different absorption coefficient, respectively[17]. Thus, 78% of light absorbance at 406 nm, which is not normally used for PDT, does not contribute to the generation of oxygen-free radicals and tumor killing. One way to compensate for this is to increase the light intensity, however, its benefit is marginal. Furthermore, increases in light intensity result in heat generation, which can damage normal healthy tissue[18–20]. Another limitation is the absence of high-throughput pipelines for the analysis of PDT outcomes[21–23]. Most existing wireless approaches utilize an RF power generator for each animal cage. Multiple RF power generators can be used, but they must be operated at least 1 m apart to avoid electromagnetic interference[24]. This is a major constraint that prohibits the high-throughput utilization of wireless methods in most laboratory settings.

Here, we introduce integrated platform that bypasses the constraints described above. It combines a DeepLabCut (DLC)[25–27]-informed low-power wireless telemetry system with a Monte Carlo (MC) thermal/light simulation. The integrated thermal/light MC simulation platform combined with a user-friendly software interface enables a bespoke PDT regimen (e.g., choice of wavelengths, determination of the number of light sources, and its placement onto an implantable device) to be delivered. The DLC-assisted low-power wireless telemetry system uses the parameters from the platform to enable the most effective PDT. The utilization of an advanced DLC algorithm for automated video analysis allows for real-time instantaneous pose estimation of the freely moving animals in a cage to ensure robust activation of animals (implanted devices) in cages. Optimized delivery of PDT via a miniaturized multichannel optoelectronic device implanted into animals allows selective and uniform delivery of multi-wavelength light to tumors. An advanced time-division multiplexing strategy, using a single RF power source, allows the activation of multiple devices/animals (<5 mice) in up to 4 animal cages, facilitating efficient use of resources.

## Results

An overview of the proposed integrated platform is shown in Fig. 1a. The platform includes: (1) integrated MC thermal/light simulation software for optimized PDT, (2) a DLC-based real-time instantaneous pose estimation algorithm for animals in a cage, and (3) a low-power wireless telemetry system for activation of a photosensitizer in a high-throughput manner. An integrated MC simulation platform leads to optimal configuration of light color and source location for effective PDT via a user-friendly software interface. Depending on the size of tumors and the photosensitizer used, it is possible to configure the light colors (wavelengths) and the number of light sources into a bespoke implantable device (Fig. 1b). DLC-based wireless telemetry uses the parameters from MC simulation to implement PDT. An advanced DLC algorithm allows for real-time instantaneous pose estimation of multiple mice (<5) in a home cage through automated video analysis. This enables the optimal control of prearranged coil-antennas to provide uniform wireless coverage and thereby the reliable activation of the devices (Fig. 1c). A time-division multiplexing strategy, paired with the DLC, permits simultaneous/independent control of multiple devices and animals (<20) in multiple cages (<4) (Fig. 1d). In Fig. 1d, two images demonstrate the activation of multiple devices in 4 cages (bottom, left) and mode of switching operation between two different configurations of light sources (bottom, middle), respectively. Videos 1 and 2 provide visual evidence of device operation.

**MC simulations for light delivery and heat management.** Figure 2a shows an overview of the integrated MC simulation platform. The platform enables numerical analysis (Monte Carlo theory) of: 1) the propagation and absorption of light (photon), and 2) heat dissipation in tissue (skin) and tumor, via a user-friendly software interface (Fig. 2a and Supplementary Fig. 1). With the selection of the photosensitizer, the number of light sources, light colors, and its formation, the integrated simulator performs the analysis. It yields the results for heat dissipation, light absorption, and delivery of light energy, allowing the most effective PDT regimen to be selected. Figure 2b–d display representative results for the photosensitizers Hypericin and Foscan, which have absorption peaks at 590 & 542 nm and 406 & 652 nm, respectively. Here, we utilize a three-dimensional model (8 cm$^3$) where a spherical object (representing the tumor volume; 4 mm in diameter) is embedded and light sources are placed on top. We also use light sources with 590 nm wavelength for Hypericin and a combination of 406 and 652 nm for Foscan. Figure 2b shows heat maps and plots of heat dissipation as a function of wavelength by constant (top, right) and duty cycled illumination (right, bottom). The results show that no detectable change in temperatures at any given parameter, limiting the damage to surrounding tissues by thermal dissipation from the light sources during device operation with 25% duty-cycle lighting. This finding is supported by temperature measurement results (Supplementary Fig. 2). Figure 2c and d summarize performance comparisons in five criteria (degree of light penetration, rate of energy absorbance, level of uniformity of energy absorbance into a tumor volume, the time required for delivery of targeted light energy, and range of temperature variation) and

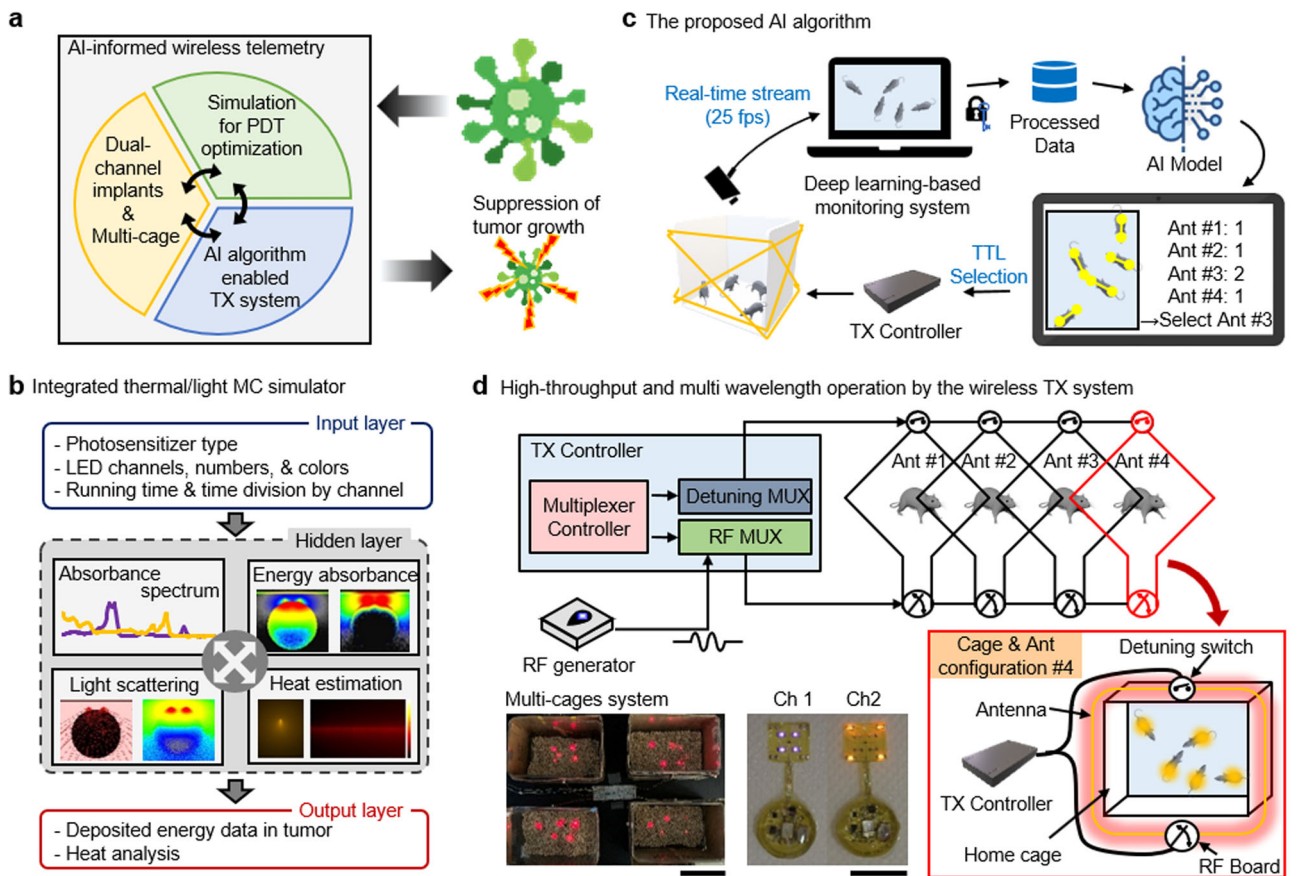

**Fig. 1 Summary of AI-enabled, implantable, multichannel wireless telemetry for PDT. a** Overview of the proposed integrated platform. **b** Workflow for an integrated MC simulator for analysis of heat dissipation and light propagation. **c** The proposed AI-based algorithm for real-time monitoring of mice. **d** Schematic illustration of the low-power wireless telemetry system for multi-wavelength operation (top). Two images represent demonstration of multichannel activation using a single power source (bottom, left) and multi-wavelength operation (bottom, right); scale bar 10 cm (left) and 1 cm (right).

show distributions of light absorption into a tumor volume for Hypericin and Foscan, respectively. Here, we set and calculate these variables from simulation in order to obtain the detailed information on the critical variables for deriving the results: (1) the capability to pass through the type of different medium, such as epidermis, dermis, and tumors, by each light source, (2) the energy absorbance ratio in the tumor according to the light spectra, and (3) the time to reach threshold energy significant for tumor growth suppression. The results, performed under different conditions (wavelength), revealed that the optimum condition for Hypericin to be most effective in PDT is reached with light sources (wavelength of 590 nm) and 25% of duty-cycle operation while that for Foscan is met with a combination of 406 and 652 nm wavelengths and 25% duty cycle operation. These were validated in in vivo and in vitro experiments. Detailed information on simulation results is found in Supplementary Figs. 3 and 4.

**Real-time identification of the optimal coil antenna based on the instantaneous pose estimation of multiple animals via DeepLabCut.** We developed software that identifies the optimal power transmission coil antenna in a real-time manner in the sense that the number of optoelectronic devices receiving wireless power is maximized. The proposed software is built on the strength of the state-of-the-art deep learning model, DLC[25–27], combined with the proposed real-time post-processing module based on the optimal graph matching algorithm, maximum weighted bipartite matching (MWBM)[28]. To be more specific, we

directly customized the DLC Python package such that the custom-trained DLC model runs with a real-time video stream. The proposed real-time post-processing module intercepts the raw estimated locations of the body parts (heads and tails) of the mice, where the estimated locations are not associated with the instance label, from the output layer, and estimates instantaneous poses (directions) of mice. Based on the estimated poses, it identifies the optimal coil antenna.

Figure 3a illustrates the procedures of the proposed software. As shown in Fig. 3a-(1), we assume that there are five freely behaving mice in which an optoelectronic device has been implanted. For the wireless power transmission, four power transmission coil antennas, only one of which activates at a time, surround the case while forming an X-shape to all sides. A webcam on the top of the cage sends a video stream to the custom-trained DLC at the rate of 25 fps. As a frame arrives, the trained DLC model detects the locations of the snouts or tails (Fig. 3a-(2)). Note that the raw output (i.e., the locations of the snouts or tails) from the output layer of DLC are not associated with the instance information. The proposed software intercepts the raw output, filters out the detected body parts of which the confidence score is less than 0.6, and passes it to the proposed real-time post-processing module. Next, the proposed post-processing module computes the matching score between all possible pairs consisting of different body parts (i.e., head-tail pair) (Fig. 3a-(3)). Based on the matching scores, MWBM finds the optimal one-to-one pairs in such a way that the sum of the matching scores of the pairs in the optimal matching set is maximized (Fig. 3a-(4)). The optimal pairs directly correspond to

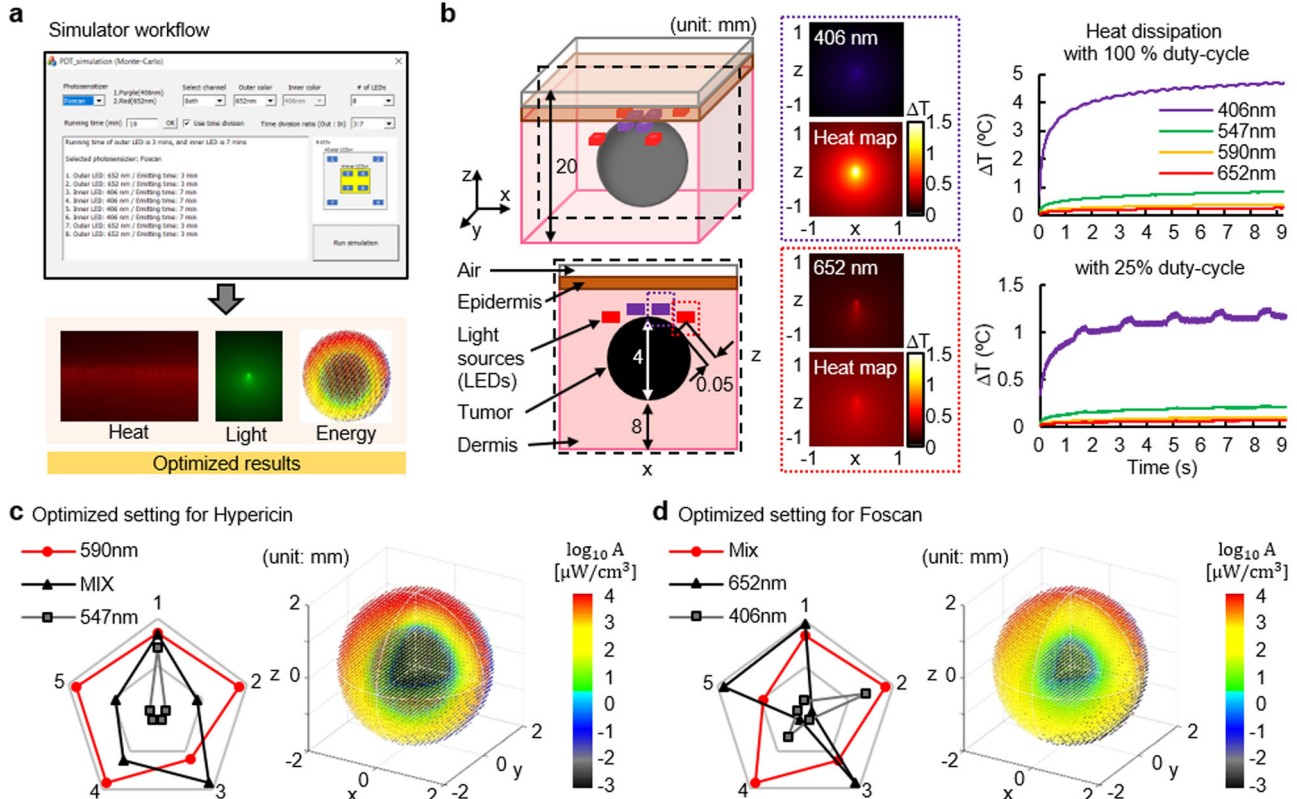

**Fig. 2 MC simulation for light delivery and heat management. a** Overview of the integrated MC simulator. The simulation platform processes parameters from a user and performs analysis of thermal dissipation, light propagation, and energy absorption. One can determine the best PDT regimen to be the most effective in activating a photosensitizer. **b** Three-dimensional and cross-sectional view of a tumor-tissue model (left). Light scattering and heatmap at a wavelength of 406 nm and 652 nm, respectively for activation of Foscan (middle). Plots of variation in temperatures as a function of time during each mode of operation; constant and 25 Hz with 10 ms on (right). The best PDT regimen for activation of Hypericin **c** and Foscan **d**. Five criteria include 1. degree of light penetration, 2. rate of energy absorbance, 3. level of uniformity of energy absorbance into a tumor cell, 4. the time required for delivery of targeted light energy, and 5. range of temperature variation.

the instantaneous poses (direction) of the mice in the cage (Fig. 3a-(5)). In turn, for each optimal pair or, equivalently, the direction of each mouse, the optimal coil antenna which is expected to achieve maximum power efficiency to each device is selected (Supplementary Fig. 5). Finally, the global optimal power transmission coil antenna is selected and activated according to the rule of the majority (Fig. 3a-(6)). The software repeats these whole procedures throughout the experiment.

For assessment of the proposed software, we use a metric, defined as the percentage of correct predictions for the data tested[29]. Figure 3b shows antenna selection accuracy for three different antenna settings: (1) two pairs of X-shaped antenna coils, one pair of X-shaped coil aligning with (2) the x- or (3) y-axis. Results revealed that the proposed software guarantees the accuracy of 80% or above in every setting that we tested (Fig. 3b). Figure 3c and d show statistics of the number of frames for two representative cases; how long a selected antenna remains activated (Fig. 3c) and how many frames (or long interval times) it takes between activation of an antenna and reactivation of itself after the first deactivation (Fig. 3d). Likely, some occupants, not all of them, in a cage may not receive enough power for activation of a photosensitizer due to a decision by the software (e.g., when two mice or vectors along the length of their body are aligned with the x-axis and corresponding vectors for the rest three mice are on the y-axis, the software chooses an antenna coil that offers better wireless coverage in the y-direction). Experimental results revealed that discharges of power stored in an embedded supercapacitor can last longer than the time intervals shown in

Supplementary Figs. 6 and 7. This suggests that the proposed software paired with the use of a supercapacitor ensures robust activation of devices in a cage. Detailed information on evaluations of the DLC software is found in Method section.

**Low-power multichannel wireless telemetry.** Figure 4a highlights signal flow from the transmission (TX) controller to antenna coils installed in each cage. The low-power wireless telemetry system employs a time-division multiplexing strategy to allow for the use of a single power source. When combined with a supercapacitor embedded in an implanted device, the proposed system enables simultaneous control of up to 4 cages. Measurement results show that an implantable device maintains constant light intensity during off cycles and thereby ensures robust activation of a photosensitizer (Fig. 4b). Most important to the multiwavelength operation is an actuation mechanism by which an individual channel is selected. Figure 4c illustrates a switching mechanism by a reed switch. When the TX system transmits a long pulse signal with an interval of a few seconds to an implantable device, a reed switch embedded in an implantable device responds to it and triggers an output of circuitry involved (Fig. 4d and Video 1). This results in reversible switching between two channels. Figure 4e shows images of a device with two different modes of switching operation. Ch1 activates 4 inner light sources in purple while Ch2 selects 4 peripheral light sources in red. Note that a reed switch enabled switching mechanism requires only tens of μA, which is a few hundred fold reductions

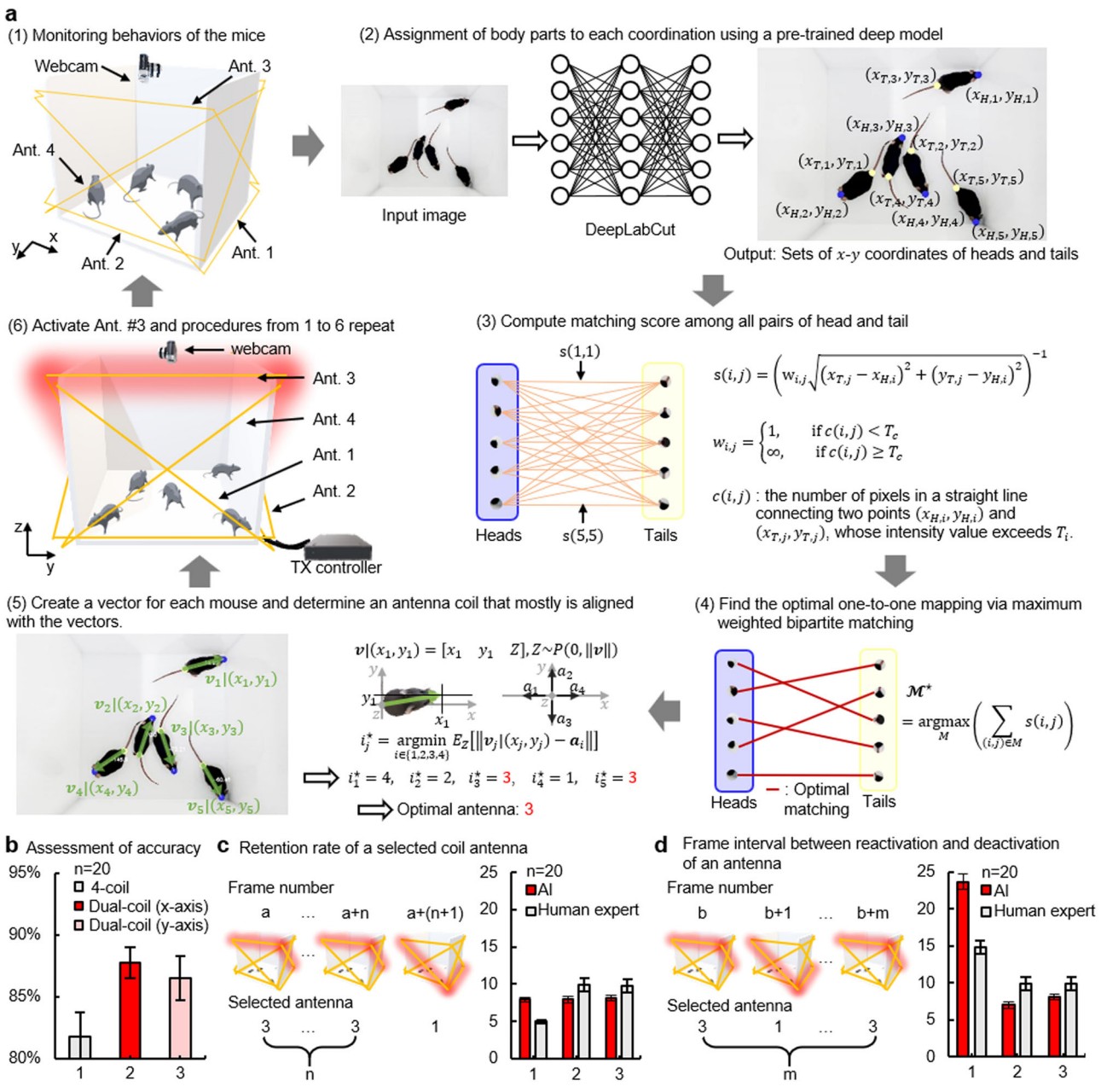

**Fig. 3 Real-time identification of the optimal coil antenna based on the pose estimation of multiple animals via DLC. a** Illustration of step-by-step procedures for the proposed algorithm. **b** Assessment of detection accuracy for three different antenna structures: (1) 4-coil (each two coil antennas are on the *x*- and the *y*-axis, respectively), (2) Dual-coil (along the *x*-axis), and (3) Dual-coil (along the *y*-axis). Statistics of the number of frames for two representative cases. **c** How long a selected antenna remains activated. **d** How many frames (how long an interval) exist between activation of an antenna and reactivation of itself after the first deactivation; Data are presented as mean values ± SD.

in power consumption compared with microcontroller employed actuation mechanisms[16,30–33]. The detailed layout and components of the dual-channel device are given in Supplementary Fig. 8, and the working principle is described in Supplementary Fig. 9.

**Hypericin and Foscan mediated in vivo/in vitro PDT.** To highlight the potential therapeutic application of the proposed DLC-based wireless telemetry system, PDT was evaluated using in vitro and in vivo in pre-clinical models of colorectal cancer. Initially, the LED surface operating temperatures were monitored over 2 days during continuous LED operation. No significant change in temperature was found over 48 h (Hour 0 vs. Hour 48; 0.3 °C increase, $p = 0.15$) (Fig. 5a). Next, Hypericin mediated PDT was performed in HT29 colorectal cancer cells, utilizing the LED devices to photoactive Hypericin. A 70% reduction in cell viability was observed between Hypericin-treated cells kept in the dark and cells irradiated with light ($p = 0.003$) (Fig. 5b). The PDT activity of the wireless LED devices was evaluated in vivo in murine models of colorectal cancer. As shown in Fig. 5c, devices were surgically implanted subcutaneously onto the right dorsal flanks of mice and positioned adjacent to HT29 tumor xenografts. Over 7 days of continuous PDT treatment, HYP(+)LED(+) treated mice demonstrated the largest tumor growth suppressing effect (Fig. 5d). By Day 7, HYP(+)LED(−) treated mice and HYP(−)

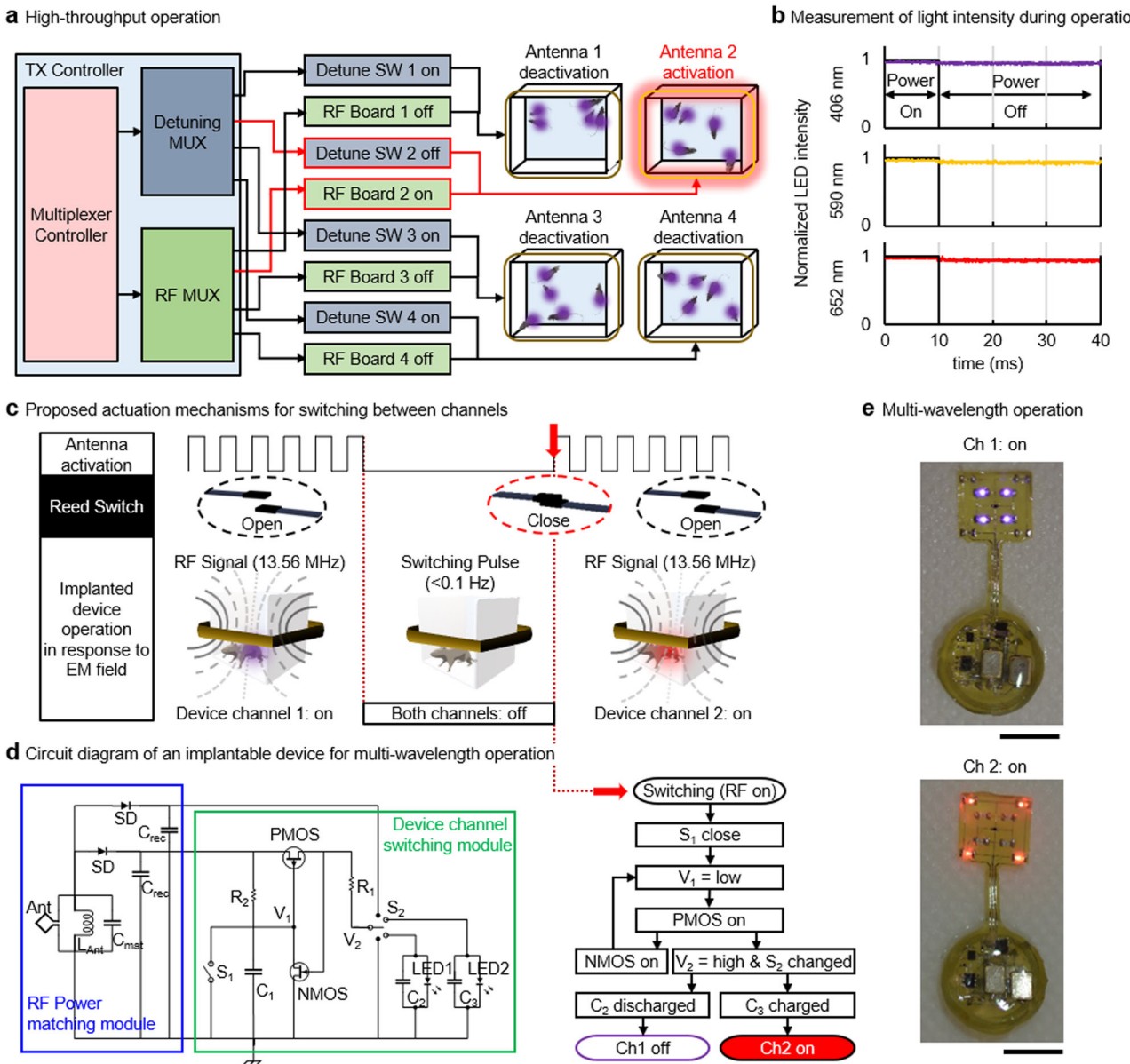

**Fig. 4 Characteristic of low-power multichannel wireless telemetry. a** Functional block diagram of multi-cage control TX system. **b** Measurements of light intensity during the TX system operation. Devices maintain constant light intensity during operation. **c** Illustration of the proposed reed switched enabled actuation mechanism. **d** Circuit diagram of an implantable device for multi-wavelength operation. **e** Images of an implantable device with multi-wavelength operation; scale bar 5 mm.

LED(+) treated mice exhibited a 2.6-fold increase ($p < 0.01$) and 6.5-fold increase ($p < 0.05$) in tumor volumes, respectively, as compared to the HYP(+)LED(+) mice. Histological analysis of HT29 tumor xenografts confirmed PDT mediated cytotoxicity in the HYP(+)LED(+) treated group, as evidenced by tumor death, which was not observed in the HYP(+)LED(−) and HYP(−) LED(+) groups (Fig. 5e). Histological analysis confirmed no systemic toxicity as evidence by histological examination of retrieved mice livers (Supplementary Fig. 10). To illustrate the advantage of combined dual-wavelength PDT over single wavelength PDT, Foscan treated HT29 cells were subjected to red LEDs light treatment or combined red/purple LEDs light treatment. Using red LEDs only, a 58% cell viability was observed, as compared to 14% cell viability in combined red/purple LEDs treated cells (Fig. 5f). To further highlight the advantage of dual-wavelength LED PDT, in vivo evaluation was conducted in murine models of HT29 tumor xenografts. After 5 days of continuous PDT treatment, a 76% decrease (combined), an 86% increase (red), a 22% decrease (purple), and 303 % increase (without LED) in tumor volumes are discovered (Fig. 5g). More information is found in Supplementary Fig. 11.

## Discussion

Most photosensitizers have more than one peak in their light absorption spectrum, with the potential to enhance photoactivation and PDT by using two or more different wavelengths. Light sources containing blue (450 nm), yellow (590 nm), and red (650 nm) LEDs have different turn-on voltages[34]. When two different wavelengths of light are connected in parallel, or share a node in a circuit, a voltage at the node becomes regulated by a LED with a lower turn-on voltage (longer wavelength of light sources) and will not reach the necessary threshold for a LED with a higher turn-on voltage

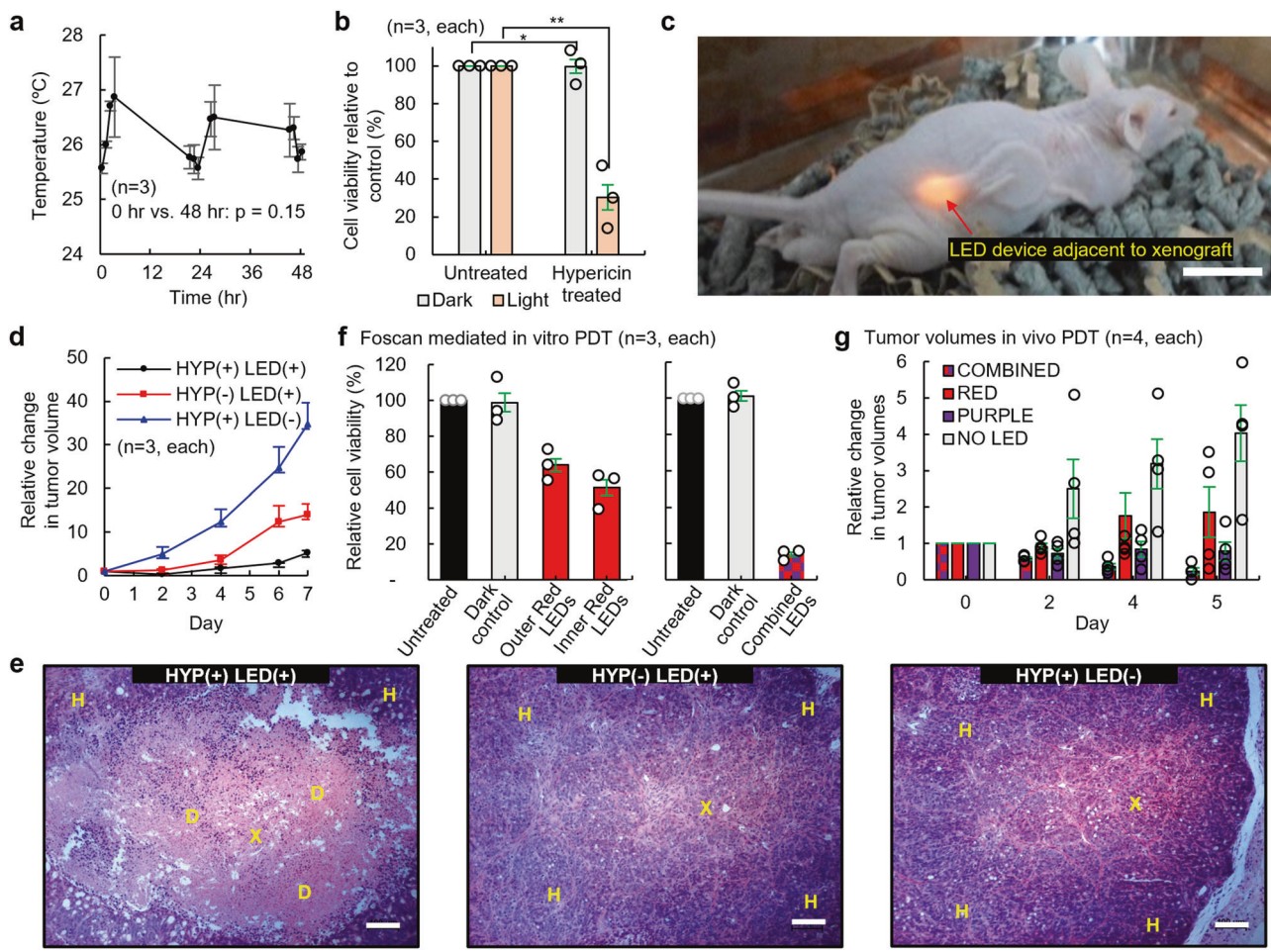

**Fig. 5 Hypericin and Foscan mediated in vitro and in vivo PDT. a** LED devices were continuously switched on and the operating temperatures of devices were measured over 48 h. **b** HT29 cells were seeded into 24-well plates and incubated with 200 nM Hypericin in the dark. LEDs were positioned under the wells and cells were treated with 36 mJ cm$^{-2}$ of light for an hour. After 24 h, cell viability was assessed. Subcutaneous HT29 tumor xenografts were generated in BALB/c nude mice and the LED devices implanted adjacent to tumor xenografts. LEDs were switched on and Hypericin was administered every 24 hours to a total of 3.5 mg kg$^{-1}$ of Hypericin and 12.1 J cm$^{-2}$ of light. **c** Photograph of a mouse with a device implanted; scale bar 1 cm. **d** Tumor volumes were recorded for a week. The HYP(−)LED(+) group received LED light treatment only. The HYP(+)LED(−) group received Hypericin treatment only. The HYP(+)LED(+) group received both light and Hypericin treatments. **e** Tumor xenografts were harvested following in vivo PDT, fixed in 4% PFA, embedded into paraffin, sectioned, and subjected to H&E staining; scale bar 100 μm. Images shown are representative examples of H&E staining performed on 3 different group. **f** HT29 cells were seeded into 24-well plates and incubated with 100 nM Foscan in the dark. Dual-channel LED devices were positioned under the wells and cells were treated with 1.8 mJ cm$^{-2}$ of light. Proposed devices contained either all Red LEDs (Red LED light treatment, left) and combined Red/Purple LEDs (30 mins outer red LEDs treatment followed by 30 mins inner purple LEDs, right): Outer denotes peripheral 4 LEDs of an implantable device and Inner indicates central 4 LEDs of it (Fig. 4e). **g** Tumor volumes were recorded for 5 days. All groups received Foscan. The combined group received both red and purple LED light treatment. The red and purple groups received red and purple LED light treatments only. No LED group received no LED light treatment. Data are presented as mean values ± SD; Two-sided unpaired *t*-test was used for statistical data analysis; *p = 0.95; **p = 0.002; X = Xenograft, H = Healthy tissue, D = Dead tissue.

(shorter wavelength of light sources). This results in activation of a single color (longer wavelength of light sources) or imbalanced light illumination. For this reason, implementation of dual- or multi-channel (or colors) in a single platform device requires an actuation mechanism for simultaneous/independent control of channels. We explored a variety of scenarios using dual color light sources, using MC simulations to extract parameters that yield the most effective output. In addition, we designed a low-power circuit to control these multichannel light sources: an actuation mechanism triggered by a reed switch enabled efficient activation of a photosensitizer with current consumption as low as tens of μA. Existing multi-channel devices are not suitable for wireless power systems because they require a stable rectified current supply of at least several tens of mA since they contain additional IC chips for multichannel control[30–32].

DLC uses deep neural networks for accurate pose estimation of user-defined animal body parts. In this work, we developed a software based on a custom-trained DLC for real-time detection of the snouts and tails of multiple mice in a video frame, where MWBM was used to match the snout and tail of each mouse. We used a matched pair of body parts (i.e., snout-tail pair) to infer the orientation of a given mouse, which can subsequently be used to optimally control the TX coil antenna for efficient wireless power delivery. While the proposed algorithm yields fairly accurate predictions, we expect that its performance may be further enhanced by incorporating a predictive model that can forecast the orientation of a given animal in the near future. Temporal sequence prediction models, such as recurrent neural networks (RNNs)[35,36], may be used for this purpose, which is currently under investigation. The potential applications also involve

quantitative analysis of complex animal behaviors such as their social interactions.

The proposed DLC-based algorithm, paired with advanced coil antennas, enables robust activation of implantable optoelectronic devices in cages. When a normal vector of an implanted device is aligned with that of the TX coil antenna, maximum wireless transmission occurs between two coils[37–39]. When misaligned, the efficiency would significantly drop. In general, reconfiguration of antennas, such as adjustment of the gap between a coil and the ground without rematching of impedance, is not recommended once they are installed in a cage[40] because transmission efficiency can significantly drop. In contrast, animals freely behave in a cage, suggesting that angles or orientations between an implanted device and the TX coil antenna vary at different times, which could be problematic. For example, two vectors become misaligned when an animal leans against a wall of a cage by standing on its hind legs or curling up. This results in a significant drop in harvesting efficiency and thereby no activation of a photosensitizer. When combined with an advanced antenna technology, and an impedance matching circuit for switching, the proposed DLC-based algorithm allows for selection from prearranged pairs of antennas, or adjustment of antenna formation, thereby leading to realignment of a TX coil antenna with implantable devices, which enables full wireless coverage in a cage.

Simultaneous activation of implanted devices in multiple animals in 4 home cages using a single RF power source is achieved primarily due to a channel isolation strategy and the use of supercapacitor. Recently, we introduced a channel isolation strategy for the operation[24]. Although it permits independent activation of up to 8 cages, it is unable to simultaneously activate the cages. A supercapacitor is crucial for the simultaneous activation of multiple cages. When a cage is activated by the TX system, a supercapacitor embedded in an implantable device in the cage can store power. When deactivated, the supercapacitor can discharge power to light sources until the cage is reselected (Fig. 4b and Supplementary Figs. 6, 7).

Construction of devices and equipment at a low cost, with easy access to the technology, could accelerate its adoption into broader scientific research exploring the mechanism of action and clinical application of PDT. The proposed optoelectronic devices consist of inexpensive commercially available components and can be built with 10 h of effort in standard cleanroom facilities. Dual-coil transmission antennas made of commercially available cheap Cu wires or tapes can be fabricated with 1 h of effort in standard laboratory facilities, and they are compatible with commercially available HF range power sources. Moreover, the wireless optoelectronic device can accommodate 8 light sources in a single platform at dimensions of 1 cm by 1 cm, with tunable wavelengths. The proposed DLC-based wireless telemetry can facilitate optimum delivery of light of PDT, tailored to the selected photosensitizer.

Although this study focused on PDT for colorectal cancer, the proposed technology is equally applicable across the spectrum of solid cancers. PDT has many advantages over conventional chemo/radiotherapy, including reduced systemic toxicity, preservation of normal tissue architecture, and avoidance of drug resistance. However, the true potential of PDT in cancer treatment has yet to be realized. In this study, the photosensitizer Foscan was chosen to elicit phototoxicity due to previous studies demonstrating great efficacy in Foscan-mediated PDT[1,17,41]. In comparison to 5-aminolevulinic acid, and its downstream intermediary product, protoporphyrin IX which tends to accumulate only in the superficial top layers of cells, Foscan is able to penetrate much deeper into solid tumors and more effectively utilize the photoactivation properties of deeply penetrating

>600 nm wavelength as well as superficial 400 nm. One of the major limitations of PDT in treating solid cancers, such as gastrointestinal cancers, has been the ability to deliver light into anatomically hard-to-reach places. The proposed solution, involving a miniaturized, biocompatible, low-power optoelectronic device to deliver light of multiple wavelengths, has the potential to open up the clinical applications of PDT. Obvious clinical applications include adjuvant therapy to treat minimal residual disease following surgical resection and the long-term palliation of cancer recurrence.

## Methods

**Ethical statement.** All the experiment protocols for Texas A&M were approved through the university. The research conducted in this study complies with all ethical regulations. All biological-based in vitro and in vivo experiment protocols were approved by the University of Leeds. The in vivo experiments were conducted within the Leeds Institute of Medical Research (University of Leeds, UK). Study was conducted in line with the Home Office regulations and in accordance with The Animals (Scientific Procedures) Act 1986, under a personal project animal license (License number: P93AOF172).

**Integrated simulator for heat dissipation and light propagation.** We combined two separate MC-based simulations (heat dissipation[42] and propagation/absorption of light[43] (photons) in tissue (skin) and tumor) and created an integrated simulation platform. Here, we used C language-based MFC library to provide a user-friendly software interface. We conducted all simulations on a computer (Intel Core i7 (7th Gen.), 8 GB RAM). The user-application guide of the developed simulation is found in Supplementary Fig. 1.

**DeepLabCut model training.** We trained the DLC model provided in the DLC Python package (version 2.2b7[25–27]) such that the trained model is capable of locating snouts and tails of up to five mice within an image (i.e., video frame). To this aim, we extracted 30 frames from a video file recorded for 10 minutes 15 s (15,375 frames in total) at the framerate of 25 frames per second via a K-means clustering algorithm to collect representative training frames. We used a ResNet-50[44,45] neural network with default parameters. For example, we optimized the network via ADAM[46] with 200,000 iterations and a gradually decreasing learning rate. As a result, the trained model achieved a validation loss of 0.0013. For all details, see config.yaml on the GitHub repository (https://github.com/parkgroup-tamu/AI-enabled-implantable-multichannel-wireless-telemetry-for-photodynamic-therapy/tree/main/DeepLabCut_Modified). We conducted all experiments including training the DLC on Lambda workstation with Intel Core i9-9960X, 128 GB RAM, and two GEFORCE RTX 2080 Ti graphics cards. All python packages used in this study are summarized in Supplementary Table 1.

**Modification of DeepLabCut Python package for identifying the optimal coil antenna.** In order to identify the optimal transmission coil antenna that maximizes instantaneous system power transmission efficiency online, we directly modified the DLC Python package (Ver. 2.2b7[25–27]). Specifically, as the original DLC python package does not support a real-time processing feature, for each frame, we intercepted the raw estimation results (i.e., the locations of snouts and tails without instance information) from the trained DLC model and used them as input to the real-time post-processing module we developed as illustrated in Fig. 3a, which was inspired by our previous study[47]. It is worth noting that there have been several attempts to add a real-time processing feature to DLC and platform it[48,49]. Although the software developed in this study was not built on these platforms to minimize potential unnecessary overhead, technically and conceptually, the modified DLC Python package has in common with these works.

As a frame arrives at the trained DLC model (Fig. 3a-(2)), the trained DLC model identifies the snouts and tails with confidence scores. In this study, we assumed that the body part whose score exceeds 0.6 is valid. As shown in Fig. 3a-(3), the proposed real-time post-processing module computes matching score $s(i,j)$ among all possible pairs of snout and tails to find the optimal pairs as follows:

$$s(i,j) = \left( w_{ij} \sqrt{\left(x_{T,j} - x_{S,i}\right)^2 + \left(y_{T,j} - y_{S,i}\right)^2} \right)^{-1}, \quad (1)$$

where $w_{ij} = \begin{cases} 1, & \text{if } c(i,j) < T_c \\ \infty, & \text{if } c(i,j) \geq T_c \end{cases}$; $x_{T,j}$ and $x_{S,i}$ are the positions of the $j$th tail and $i$th snout on the $x$-axis, respectively; and $y_{T,j}$ and $y_{S,i}$ are the positions of the $j$th tail and $i$th snout on the $y$-axis, respectively. $c(i,j)$ is the number of pixels, whose intensity value is greater than or equal to $T_c$, on the straight line connecting two points $(x_{S,i}, y_{S,i})$ and $(x_{T,j}, y_{T,j})$. Next, the proposed module finds the optimal one-

to-one mapping set $\mathcal{M}^*$ via the MWBM as follows (Fig. 3a-(4)):

$$\mathcal{M}^* = argmax_M \left( \sum_{(i,j) \in M} s(i,j) \right). \quad (2)$$

Construction of the optimal one-to-one mapping set $\mathcal{M}^*$ naturally leads to identifying the projection vector $\boldsymbol{v}_j|(x_j, y_j)$ of the orientation vector $\boldsymbol{v}_j$ on the $xy$-plane of the $j$th mouse. Under the practically reasonable assumption that a mouse can only change the angle up to 90 degrees in the direction the mouse are looking (i.e., mice do not stand on their hands), the optimal antenna index $i_j^*$ for $j$th mouse can be found as follows:

$$i_j^* = argmin_{i \in \{1,2,3,4\}} E_Z[\|\boldsymbol{v}_j|(x_j, y_j) - \boldsymbol{a}_i\|] \quad (3)$$

where $\boldsymbol{a}_i$, $i = 1, 2, 3, 4$, is the antenna vector of the $i$th coil antenna (See supplementary Fig. 5). Finally, the optimal coil antenna that maximize the instantaneous system power transmission efficiency is determined by the principle of majority vote.

**Quantitative performance assessment of the instantaneous pose estimation algorithm for identifying the optimal transmission coil antenna.** The proposed DLC-based algorithm yields the following information for each frame: (1) the position of the snout and tail of each mouse, (2) the direction in which each mouse is heading toward, and (3) the angle between a vector along the length of each mouse and the $y$-axis. Based on this information, the algorithm selects an antenna coil that leads to the best wireless coverage in a cage. The following are three antenna settings considered in this study: (1) Two pairs of X-shaped coil antenna, X-shaped coil antenna in (2) the $x$-axis direction and (3) the $y$-axis direction. For the quantitative performance assessment of the algorithm, we used three video recordings, each of which is 10 minutes running time. We randomly extracted and evaluated 20 frames from a total of 15,000 frames in each video recording and repeated the procedures twenty times. For each set of 20 frames, we compared the decision made by the proposed algorithm for each frame in the given set with the one made by a human expert in each antenna setup. Supplementary Fig. 12 shows a representative example of an image (video frame) processed by the algorithm. To check the performance of each antenna setup and implanted devices, we focused on the following two statistics (in terms of the number of frames): how long a selected antenna remains activated (Fig. 3c) and how many frames (i.e., how long of a time interval) it takes between activation of an antenna and its reactivation after the first deactivation (Fig. 3d). Here, a human expert extracted and analyzed the data, which had been processed by the proposed algorithm, in every 20 frames. For Fig. 3c, we chose a mouse (implanted device) from the group and measured how long a selected antenna remains activated or aligned with a vector determined by the mouse as a function of frames. Similarly, for Fig. 3d, we measured a time interval as a function of the number of frames between deactivation of an antenna and subsequent reactivation of it. These were averaged for 20 trials, leading to the statistics shown in Fig. 3b–d.

**Device fabrication.** The pattern fabrication process began to mount a flexible copper/polyimide (Cu/PI) bilayer film (thickness; 12 μm/18 μm, AC181200RY, Dupont™ Pyralux®) onto a glass slide (dimensions, 5.08 cm by 7.62 cm). In the cleanroom facility, we deposited the photoresistor onto the Cu layer for 2 μm thick (AZ 1518, AZ®, recipe; spin-coated at 4000 r.p.m. for 20 s) and illuminated UV lights to lithograph patterns for pads and interconnections (EVG610, EV Group, recipe; UV intensity for 100 mJ cm⁻²). To engrave the photoresistor layer, the Cu/PI film was immersed in developer solution (AZ Developer 1:1, AZ®) for 30 s and washed with distilled water for 10 s. Next, immersion in copper etchant (LOT: Z03E099, Alfa Aesar™) for 10 min and rinses with solvents: acetone, methanol, and isopropanol in order, and distilled water for 1 min defined Cu patterns such as interconnections and pads on the flexible PI layer. In the standard laboratory facility, chip components including SMD (surface-mount-device) LEDs, passive components, and IC components were mounted onto the pattern using a soldering machine. For encapsulations, we applied Polydimethylsiloxane (PDMS) (Sylgard™ 184 silicone elastomer kit, Dow®; 10:1 mix ratio) with a dip-coating process (500 μm thick) to the sample, and then it was cured in the oven at 80 °C for 1 h. These procedures yield a proposed multi-wavelength optoelectronic implant.

**Antenna-coil fabrication and wireless, power-control system.** We worked with 8-ga bare Cu wire for the sub-antenna coil and Cu tapes (0.635 mm thick by 2.54 cm wide) for the source-antenna coil. The sub-coils were placed under a cage and around all sides of the cage while the source-coil was situated in the center of the crossed-long side of the cage vertically. Impedance matching using Network Analyzer (ENA Series E5063A, Keysight) with a discrete capacitor component yielded source and sub-coils, each of which resonates at 13.56 MHz (the source coil) and 15 MHz (the sub-coils), respectively. Wireless power control systems consisted of a RF power generator (ID ISC.LRM2500-A, FEIG Electronics), and an auto-tunable matching board (ID ISC.DAT-A, FEIG Electronics). For the multi-cage system, it requires a TX controller including an RF multiplexer (ID ISC.ANT.MUX.M8, FEIG Electronics), a control board (nRF52832 Development Kit, Nordic semiconductor), and a customized decoupling multiplexer.

**Finite element-methods analysis.** For numerical electromagnetic simulations of the proposed antenna structure, we used a finite element-methods analysis tool (Ansys Electromagnetics Suite 17-HFSS, Ansys®) to compute distributions of the electromagnetic field in a home cage. Antenna coils made of copper stripes or wires were modeled to materials with finite conductivity, 58 MS s⁻¹. We figured out the residual dependency of transmitted power on angles and orientations between an implanted device and the TX coil antenna in the experimental box. All simulations were conducted with a TX level of 4 W, which is far below guidelines suggested by IEEE or ICNIRP[50,51] (Supplementary Fig. 13).

**Optical and thermal characteristics of the implant device.** The residue light intensity, which keeps in response to the capacitance after the cutoff of the power source, was measured using a photodiode and oscilloscope. This was conducted repeatedly in three colored LEDs that have different turn-on voltages. For thermal assessments of wireless devices, we measured heat dissipation of the light sources in a device using an infrared camera (VarioCAM HDx head 600, InfraTech) in two different conditions: a device installed under a sealed bag of saline solution (10% PBS) instead of a mouse, and a device itself in the cage. The power supply was a function of time at duty cycles of 25% with a 10 ms pulse train, which is the same as experimental conditions by the wireless TX system.

**Materials for PDT in colorectal cancer models.** The human colorectal adeno-carcinoma cell line, HT29, was obtained from the European Collection of Authenticated Cell Cultures (Salisbury, UK) and cultured in Roswell Park Memorial Institute (RPMI) 1640 Medium plus GlutaMAX™ (Gibco® by Life Technologies™, Paisley, UK) supplemented with 10% (v/v) Foetal Bovine Serum (FBS) (Sigma-Aldrich, Gillingham, UK). The photosensitizers Hypericin and Foscan were obtained from Sigma Aldrich and biolitec Pharma Ltd. (Jena, Germany) respectively. Thiazolyl Blue Tetrazolium Bromide (MTT) was obtained from Sigma Aldrich. Hypericin and Foscan stock solutions were prepared in 100% ethanol. Final concentration of ethanol in cell media for Hypericin was 0.2% for in vitro experiments and 6% ethanol per 0.5 mg kg⁻¹ injection. Final concentration of ethanol in cell media for Foscan was 0.5% for in vitro experiments and 10% ethanol per 0.5 mg kg⁻¹ injection.

**Monitoring implantable device operating temperatures.** Implantable LED devices were switched on at room temperature and allowed to continuously run for 48 h. The surface operating temperature of the miniature LEDs was measured over 48 h using an RS PRO medical infrared thermometer (RS Components Ltd., Corby, UK).

**In vitro PDT cytotoxicity.** HT29 cells were seeded into 24-well tissue culture plates (Corning Inc., New York, USA) at 2 ×10⁵ cells per well and incubated at 37 °C/5% CO₂/95% for 24 h. Cells were treated with 200 nM Hypericin or 100 nM Foscan in the dark for 16 h. Cell cultures were then washed with Dulbecco's Phosphate-Buffered Saline (DPBS, Gibco® by Life Technologies™) and Phenol red-free RPMI 1640 medium with L-glutamine (Gibco® by Life Technologies™) supplemented with 10% (v/v) FBS was added to cultures. LED devices were positioned in the center and underneath the wells and switched on. For single-channel LED devices, light treatment lasted for 1 h at 10 μW cm⁻², equating to 36 mJ cm⁻² of total light dose. For dual-channel LED devices, light treatment lasted for 1 h at 0.5 μW cm⁻², equating to 1.8 mJ cm⁻² of total light dose. Depending on the experimental conditions, cultures were either irradiated with light or kept in the dark at room temperature. After 24 hours, an MTT cell viability assay was performed by dissolving Thiazolyl Blue Tetrazolium Bromide into Phenol red-free RPMI 1640 medium (MTT solution). Cell media was discarded and replaced with MTT solution and cultures were incubated in the dark for 3 h. The MTT solution was discarded and formazan crystals were dissolved using propan-1-ol. Optical absorbance values were measured using a Mithras LB 940 Microplate Reader (Ex: 570 nm) (Berthold Technologies Ltd., Harpenden, UK).

**In vivo metronomic PDT.** Animal experiment was performed at the University of Leeds under a personal project home office license held by P. Louise Coletta. The in vivo experiments were conducted within the Leeds Institute of Medical Research (University of Leeds, UK). Study was conducted in line with the Home Office regulations and in accordance with The Animals (Scientific Procedures) Act 1986, under a personal project animal license (License number: P93AOF172). Nine female BALB/c nude mice (6–8 weeks old) were used for the Hypericin-mediated in vivo metronomic PDT experiment. Mice were purchased from Charles River UK, Ltd (Margate, UK). One million ($1 \times 10^6$) HT29 cells were suspended in 100 μL of FBS-free RPMI and injected subcutaneously into the right dorsal flank area of mice. Inoculated HT29 cells were grown for 8 days to generate heterotopic HT29 colorectal cancer tumor xenografts. Following the growth of tumor xeno-grafts, the miniature implantable LED devices were sterilized in 70% ethanol and surgically implanted into mice. The LED-containing probes were positioned adjacent to tumor xenografts. 3 M™ Vetbond™ Tissue Adhesive surgical glue (3 M™ United Kingdom PLC., Bracknell, UK) was used to close the wounds (Supplementary Fig. 14).

Immediately following device implantation (Day 0), mice were intraperitoneally injected with 0.5 mg kg$^{-1}$ Hypericin prepared as working solutions in DPBS, and LEDs were switched on. 0.5 mg kg$^{-1}$ Hypericin injections were administered daily. Tumor xenograft volumes were measured and recorded on Days 0, 2, 4, 6, and 7. The HYP(−)LED(+) group received LED light treatment only. The HYP(+)LED(−) group received Hypericin treatment only. The HYP(+)LED(+) group received both light and Hypericin treatments. Over 7 days, mice received total doses of 3.5 mg kg$^{-1}$ Hypericin (0.5 mg kg$^{-1}$ Hypericin per day) and 12.1 J cm$^{-2}$ of light (light fluency rate = 20 μW cm$^{-2}$). Following completion of the experiment, mice were euthanized in accordance with Schedule 1 of the Animals (Scientific Procedures) Act 1986 and the tumor xenografts and mice livers were harvested.

For the Foscan-mediated in vivo PDT experiment, twenty female BALB/c nude mice (6–8 weeks old) were used. Mice were implanted with miniature LED devices on Day 0 and injected intraperitoneally with 0.5 mg kg$^{-1}$ Foscan, prepared as working solutions in DPBS, and LEDs were switched on. 0.5 mg kg$^{-1}$ Foscan intraperitoneal injections were administered daily. Tumor xenograft volumes were measured and recorded on Days 0, 2, 4 and 5. All groups received daily Foscan injections. The combined group received both red and purple LED light treatment. The red group received red LED light treatment only. The purple group received purple LED light treatment only. The no LED group received no LED light treatment. Over 5 days, mice received total doses of 2.5 mg kg$^{-1}$ Foscan (0.5 mg kg$^{-1}$ Foscan per day) and 0.43 J cm$^{-2}$ of light (light fluency rate=1 μW cm$^{-2}$). Following the completion of the experiment, mice were euthanized in accordance with Schedule 1 of the Animals (Scientific Procedures) Act 1986.

For the animal experiments, mice were housed in the GM500 Mouse IVC Green Line cage (Tecniplast UK, London, UK) and kept on the Sealsafe Plus Mouse DGM cage rack (Tecniplast UK). Mice were kept on a 12-hour light and 12-hour dark cycle, at 25 °C room temperature and ambient room humidity. The maximal tumor size, as permitted under the Home Office regulations and the University of Leeds review board is 15 mm tumor xenograft diameter in any one direction. The maximal tumor size was not exceeded in animal experiments.

**Histological analysis of tissue**. Harvested tumor xenografts and livers were fixed in 4% (w/v) paraformaldehyde (PFA) for 24 h and stored in 70% (v/v) ethanol at 4 °C. Fixed tissue was embedded in paraffin, sectioned onto glass slides, and subjected to Haematoxylin and Eosin (H&E) staining. Stained slides were imaged using the Nikon Eclipse E1000 Microscope (Nikon UK Ltd, Kingston upon Thames, UK).

**Statistical analysis**. The unpaired two-tailed Student's $t$-test was used to perform statistical analysis using GraphPad Prism 9 (GraphPad Software, Inc., California, USA). $p < 0.05$ was considered to be statistically significant. Data are presented as the mean ± standard deviation.

**Reporting summary**. Further information on research design is available in the Nature Research Reporting Summary linked to this article.

## Data availability
The data supporting the results in this study are available within the paper and its Supplementary Information. Source data are provided with this paper.

## Code availability
The code and the trained DLC model are available from GitHub (https://github.com/parkgroup-tamu/AI-enabled-implantable-multichannel-wireless-telemetry-for-photodynamic-therapy/tree/main/DeepLabCut_Modified).

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

## Acknowledgements

This work was supported by grants from the interdisciplinary X-Grants Program, part of the President's Excellence Fund at Texas A&M University (S. Park, B. Yoon), 2018 NARSARD Young Investigator Awards (S. Park) from Brain & Behavior Research Foundation and National Science Foundation Engineering Research Center for Precise Advanced Technologies and Health Systems for Underserved Populations PATHS-UP (EEC-1648451; S. Park). This work was also supported by a Wellcome Trust Institutional Strategic Support Fund Fellowship (204825/Z/16/Z; M. I. Khot), a National Institute for Health Research (NIHR) Research Professorship (D. G. Jayne), an NIHR Senior Investigator Award (D. G. Jayne) and the NIHR infrastructure at Leeds. The views expressed are those of the authors and not necessarily those of the National Health Service, the NIHR, or the Department of Health. We would like to thank Syed Khawar Abbas (University of Leeds) for performing the surgical implanting procedures. S. Park would like to express thanks to Mr. Woonki Park and Dr. Gerald Coté (Texas A&M University) for general advice.

## Author contributions

W.K. designed a dual-channel device and multi-cage transmission system, fabricated devices and antennas, developed and integrated the MC simulation, tested devices and antenna, and generated figures. M.I.K. conducted the in vitro and in vivo experiments, analyzed results, and generated figures, H.W. developed the DLC algorithm, and generated figures. S.H. conducted circuit and electromagnetic field simulation, developed the heat dissipation simulation, fabricated devices and antennas, tested devices. D.B. designed the device prototype. T.M. conducted the in vitro and in vivo experiments and histological analyses. B.D. verified the DLC software performance. P.C. oversaw the animal experiments. D.G.J. provided resources and edited the manuscript. B.Y. oversaw the development of the AI algorithm and its performance analysis and edited the manuscript. S.P. provided resources and edited the manuscript. W.K., M.I.K., H.W. and S.P. wrote the manuscript. S.P. oversaw all experiments and data analysis.

## Competing interests

The authors declare no competing interests.
