## [Peer Review File · Nature Communications]

Reviewers' Comments:

Reviewer #1:

Remarks to the Author:

This is an interesting but rather confusing paper. They have mixed several elements together: Monte Carlo simulations of two different wavelengths; implantable wireless powered LED devices; AI control of antennae coils by video tracking of mice; metronomic PDT.

1. In my opinion, if these three elements had been addressed one by one, it would have been less confusing
2. The abstract gives the impression that both hypericin and Foscan were used in mice, but in reality Foscan was only used in vitro. Since the advantage of using dual wavelength (blue and red) is much more pronounced for Foscan, this should have been tested in vivo
3. One justification is mistaken. It is well known that a moderate increase in temperature (hyperthermia) potentiates the effects of PDT, so major efforts to restrict heat production are misguided
4. One of the best justified applications of metronomic PDT is ALA-induced PPIX because ALA can be administered in drinking water. This should have at least been discussed as the blue and red light excitation would have been ideal
5. In fig 5g "inner and outer" LEDs are confusing
6. Hypericin is not soluble in DPBS so it must have contained some % of ethanol
7. You need to give the IACUC approval number for the protocol
8. There are many mistakes, I shall try to list some of them
9. Singlet oxygen is not an "oxygen free radical"
10. Absorption coefficient not "absorption rate"
11. "EM waves at UHF ranges (300 MHz ~ 3 GHz) are not transparent to EM waves," is meaningless
12. You cannot activate an animal only a device
13. A tumor cell cannot be 4mm in diameter

Reviewer #2:

Remarks to the Author:

In this manuscript, the authors presented AI-informed low-power wireless telemetry with an integrated thermal/light simulation platform for photodynamic therapy (PDT). The AI-assisted wireless telemetry enables adequate illumination of tumors through high-throughput (<20 mice) and multi-wavelength operation. Furthermore, Hypericin and Foscan mediated PDT demonstrates the significant suppression of tumor growth.

The AI-assisted wireless telemetry combined with a supercapacitor-enabled switching mechanism allows for continuous and efficient delivery of multichannel light to mice in a cage and to multiple cages, potentially enabling more efficient and high throughput PDT study in animals. This highly multidisciplinary work is novel and can be of great interest to others in the community. However, some major revisions are needed to clarify and strengthen some of the conclusions.

Major:

1. The motivation/introduction part is somewhat confusing and should be clarified. For example, in the abstract, the authors mentioned the challenge is "Existing approaches... suffer from poor spatiotemporal resolution due to inability to minimize oxygen depletion in a tumor." It is not clear why poor spatiotemporal resolution is caused by oxygen depletion – they seem to be two parallel goals. In addition, in the first paragraph, it was mentioned that the current PDT reached a plateau due to poor spatiotemporal resolution. In the second paragraph, it was pointed out that recent advances in wireless technologies improved the spatiotemporal resolution, but the challenges are compensation for light loss versus tissue heat/damage by high intensity light and UHF EM waves. It is somewhat confusing what the proposed technology is trying to achieve – improve the spatiotemporal resolution, minimize oxygen depletion, or reduce tissue heat/damage by using multichannel light sources?

2. Following the previous comment, if one of the major advantages of this technology is to reduce oxygen depletion, some evidence showing that this new method could reduce oxygen depletion in tumors should be provided to support the claim. If the focus is on the capability of delivery light at multiple wavelengths continuously at a low power and the activation of mice in multiple cages, the abstract and the introduction should be modified accordingly.

3. Measurement of heat dissipation using IR camera (Supplementary Figure 2): in this figure, the devices are kept inside a bag filled with saline solution, and the IR camera is measuring the temperature from the top. It seems that the camera would be measuring the average temperature of the solution or the bag which may not be the temperature right at or next to the device.

4. The in vivo experiments compared HYP(+)/LED(+) conditions with controls. However, the multichannel (multiple wavelength) work has only been demonstrated in vitro. Since the multichannel capability is a major contribution of this work, an in vivo demonstration of the multichannel/wavelength activation would be very helpful. Is it realistic to perform in vivo demonstration of the multichannel activation? How might the movement of the implant affect the results?

Minor:

5. On page 5, the authors use the five criteria (1-5) to choose the optimal condition for PDT. Some sentence describing how these 5 criteria relate to the optimum condition should be added. In addition, "Results performed under different conditions (wavelength) revealed that uniform light delivery and minimum heat dissipation for Hypericin are reached with..." Uniform light delivery and minimum heat dissipation only refers to criteria 3 and 5. A more comprehensive description may also include amount/rate of energy absorption, etc.

6. In figure 2b, what is the distance between the LED and the tumor? Is it chosen based on the experimental conditions during the in vitro testing (such as the condition in Fig.5g)?

7. On page 10, "...power budgets as low as tens of μ A." Why does the power have a unit of μ A? Can the authors estimate the power consumption with the reed switch compared with the power consumption simply using multiple light sources?

Reviewer #3:

Remarks to the Author:

I am not a domain expert in the main subject matter of this paper. I am, however, an expert in animal pose estimation. Thus, I will only comment on these aspects. The combination of antenna tracking and deeplabcut is a nice contribution (and from a non-expert view, the paper is quite exciting). However, I have questions about the AI method being used.

It appears the authors are using a beta release of DeepLabCut code that has not been published (or no preprint yet). Therefore, performance is quite unclear for this method. There are also not sufficient details in the method to understand what the authors used exactly. In the main text they mention using neural networks for pose estimation in multiple animals, and then in the methods they say deeplabcut 2.2b7-- looking at this code it's clear there are many options that are not clearly described in the methods. Moreover, the authors mention bipartite matching but the methods don't detail this fully. It seems unusual, to me, that the authors here are using unpublished code without proper benchmarking and performance metrics. I give more concrete examples below.

"Modification to DLC model

We utilized, custom-trained, and modified the DLC Python package (Ver. 2.2b7). Specifically, we used the custom-trained DLC model to estimate the locations of the body parts such as snouts and tails of the mice within an image (i.e., video frame). Note that the original DLC python package does not support a real-time processing feature, instead it only runs on video files. Hence, we directly modified the Python package in such a way that it can infer the locations of the body parts

of the mice and estimate the optimal coil antenna through the functional modules, illustrated in Fig. 3a, in a real-time manner. We conducted all experiments including training the DLC on a GPU workstation (Lambda workstation with Intel Core i9-9960X, 128 GB RAM, and two GEFORCE RTX 2080 Ti graphics cards)."

- It appears the authors are using unpublished DeepLabCut code: 2.2beta7 is a beta release. It would seem appropriate to ask those authors how to appropriately cite this version? Specifically, Figure 3A 2-4 seems to be closely related to the unpublished multi-animal DeepLabCut contribution. DLC has 14 networks available, multi-data augmentation steps, new steps introduced in 2.2.beta, etc. The authors should give sufficient details to be clear to the reader what they used.

- There are also several papers that present real-time options for DeepLabCut; can the authors comment on the differences here from Forsys et al, Kane et al, etc? (Technically, DLC could always take in frames, whether in a video or frame format-- it is a frame-by-frame pose estimation algorithm that always allowed for batch size 1). Note that in Kane et al, 2020 there is also a novel predictive model to reduce delays in low-latency tracking.

- The authors should be sure to cite the open source code they use, such as Matplotlib, numpy, and importantly the DLC Python package of the code (Nath et al, 2019).

Minor Points:

"The utilization of an advanced AI algorithm for automated video analysis allows for real time tracking of the freely moving animals in a cage to ensure robust activation of animals (implantable devices) in cages"

-- the authors might consider stating what they used, vs. the flashy and non-informative "AI"; simply stating they used DeepLabCut is sufficient.

Figure 3. (a) Illustration of step-by-step procedures for the proposed artificial intelligence algorithm.

-- this is not a newly "proposed" software (this is DeepLabCut), so please consider revisiting the figure caption and also be sure to include a [™] to the DLC image you have in Figure 3.

Reviewer #1 (Remarks to the Author):

Comments:

This is an interesting but rather confusing paper. They have mixed several elements together: Monte Carlo simulations of two different wavelengths; implantable wireless powered LED devices; AI control of antennae coils by video tracking of mice; metronomic PDT.

1. In my opinion, if these three elements had been addressed one by one, it would have been less confusing.

Response: The reviewer's comments are well taken. We presented the results section sequentially as the reviewer commented: 1) overview, 2) MC simulations, 3) DeepLabCut-enabled antenna control system by video tracking of mice, 4) low-power implantable wireless powered LED devices, and 5) metronomic PDT. However, the abstract and introduction sections tended to be a bit confusing. So, we revised the Abstract and Introduction to the following:

"Advances in wireless technologies enable remote delivery of light to tumors, but suffer from key limitations, including low levels of tissue penetration and PS activation." & "Together, they establish a range of guidelines for effective PDT regimen design. *In vivo* Hypericin and Foscan mediated PDT, using cancer xenograft models, demonstrated substantial suppression of tumor growth, warranting further investigation in research and/or clinical settings." in Abstract, and "Existing hardware PDT, in particular the light source, suffers from key limitations in their ability to: 1) deliver light to deep tumor cells (e.g., physical tethers to light sources would complicate access to targeted tumor cell), 2) control light sources and/or wavelengths in a programmed manner (e.g., light sources with single wavelength would activate a photosensitizer (PS) at a modest rate if a PS has two absorption peaks), and 3) be multiplexed so that multiple animals can be studied in parallel.", in 2nd paragraph of Introduction; "For example, Foscan has two absorption peaks at 406 nm and 652 nm with different absorption coefficients¹⁷. Thus, 78% of light absorbance at 406 nm, which is not normally used for PDT, does not contribute to the generation of oxygen-free radicals and tumor killing. One way to compensate for this is to increase the light intensity, however, its benefit is marginal. Furthermore, increases in light intensity result in heat generation, which can damage normal healthy tissue¹⁸⁻²⁰. Another limitation is the absence of high-throughput pipelines for the analysis of PDT outcomes²¹⁻²³. Most existing wireless approaches utilize an RF power generator for each animal cage. Multiple RF power generators can be used, but they must be operated at least 1m apart to avoid electromagnetic interference²⁴. This is a major constraint that prohibits the high-throughput utilization of wireless methods in most laboratory settings." in 3rd paragraph of Introduction.

2. The abstract gives the impression that both hypericin and Foscan were used in mice, but in reality Foscan was only used in vitro. Since the advantage of using dual wavelength (blue and red) is much more pronounced for Foscan, this should have been tested in vivo.

Response: We agree with reviewer's comment. We conducted animal experiments of each three group with Foscan: 1) red LEDs only, 2) purple LEDs only, and 3) combined dual red/purple colors in a device. As we expected, the tumor's growth was dramatically suppressed by dual color device (Figure 5g). We added the following sentence in the last paragraph of Result 4th subheading section: "To further highlight the advantage of dual-wavelength LED PDT, *in vivo* evaluation was conducted in murine models of HT29 tumor xenografts. After 5 days of continuous PDT treatment, a 0.89-fold (89 %) decrease, a 0.51-fold (51 %) decrease, a 0.64-fold (64 %) decrease, and 1.94-fold (194 %) increase in tumor volumes (Fig. 5g)". We have included data in Figure 5.

3. One justification is mistaken. It is well known that a moderate increase in temperature (hyperthermia) potentiates the effects of PDT, so major efforts to restrict heat production are misguided.

Response: The purpose of monitoring temperature was to ensure that there was no increase in temperature during LED treatment. An increase in LED operating temperatures, even moderate, could potentially lead to adverse toxicity in normal tissue or impact the overall wellbeing of animals, which would severely limit the impact of PDT. In this study, we were interested in purely the phototoxicity effects of PDT when light sources affixing to the tumor only without affecting other healthy tissue.

4. One of the best justified applications of metronomic PDT is ALA-induced PPIX because ALA can be administered in drinking water. This should have at least been discussed as the blue and red light excitation would have been ideal.

Response: PPIX has previously been shown to be a good photosensitizer for superficial lesions and tumours, where the biggest impact is seen only in the outer layers of cells. This is unlike Foscan, that can penetrate deeper in tumours and studies have shown this drug to exert phototoxic activities in deeper layers of cells and has a more penetrative effect. We added the following sentence in the last paragraph of Discussion section: "In this study, the photosensitizer Foscan was chosen to elicit phototoxicity due to previous studies demonstrating great efficacy in Foscan-mediated PDT. In comparison to 5-aminolevulinic acid, and its downstream intermediary product, protoporphyrin IX which tends to accumulate only in the superficial top layers of cells, Foscan is able to penetrate much deeper into solid tumours and more effectively utilize the photoactivation properties of deeply penetrating >600nm wavelength as well as superficial 400nm."

5. In fig 5g "inner and outer" LEDs are confusing.

Response: Reviewer's comment is well taken. We changed the term in Figure 5(g) and add the following comments in Figure 5 legend: "Peripheral denotes outer 4 LEDs of an implantable device and Central indicates inner 4 LEDs of an implantable device; images of an implantable device in Fig4(e)".

6. Hypericin is not soluble in DPBS so it must have contained some % of ethanol.

Response: Both Hypericin and Foscan stocks in 100% ethanol. Final concentration of ethanol in cell media was 0.2% for *in vitro* experiments and 6% ethanol per 0.5mg/kg injection for Hypericin and 0.5% for *in vitro* experiments and 10% ethanol per 0.5mg/kg injection for Foscan. No adverse toxicities was observed in any mice in all treatment groups. We added the following sentences in Methods-Materials for PDT in colorectal cancer models: "Hypericin and Foscan stock solutions were prepared in 100% ethanol. Final concentration of ethanol in cell media for Hypericin was 0.2% for *in vitro* experiments and 6% ethanol per 0.5mg/kg injection. Final concentration of ethanol in cell media for Foscan was 0.5% for *in vitro* experiments and 10% ethanol per 0.5mg/kg injection."

7. You need to give the IACUC approval number for the protocol.

Response: The *in vivo* experiments were conducted in within the Leeds Institute of Medical Research (University of Leeds, UK). Study was conducted in line with the Home Office regulations and in accordance with The Animals (Scientific Procedures) Act 1986, under a personal animal licence (Licence number: P93AOF172). We modified the following sentences in Methods-In vivo metronomic PDT: "The *in vivo* experiments were conducted within the Leeds Institute of Medical Research (University of Leeds, UK). Study was conducted in line with the Home Office regulations and in accordance with The Animals (Scientific Procedures) Act 1986, under a personal animal licence (Licence number: P93AOF172)."

8. There are many mistakes, I shall try to list some of them.

9. Singlet oxygen is not an "oxygen free radical".

Response: Oxygen free radical has been changed to be singlet oxygen. We modified the following sentences in line 68: "For example, tissue oxygenation in tumors is critical for the production of singlet oxygen and successful phototoxicity, with tumor hypoxia being a limiting factor to PDT efficacy PDT^{6,7}."

10. Absorption coefficient not "absorption rate".

Response: We agree with reviewer's point. We modified the following sentence in line 84-85: "For example, Foscan has two peaks at 406 nm and 652 nm in absorption spectra according to different absorption coefficient, respectively."

11. "EM waves at UHF ranges (300 MHz ~ 3 GHz) are not transparent to EM waves," is meaningless.

Response: We removed the sentence from Introduction.

12. You cannot activate an animal only a device.

Response: We mentioned a sentence, “simultaneous activation of multiple animals in 4 home cages using...” in line 285. Here, an animal indicates not exactly animal itself, but it means an animal which is implanted a device fully onto the right dorsal flank area already. Like reviewer’s comment, it seems to occur misunderstandings, so we modified the following sentence:

“Simultaneous activation of implanted devices in multiple animals in 4 home cages using a single RF power source is achieved primarily due to a channel isolation strategy and the use of supercapacitor.”

13. A tumor cell cannot be 4mm in diameter.

Response: We correct it as tumor volume in the 2nd paragraph of Results section: “(representing the tumor volume; 4 mm in diameter)”, and “Figs. 2c and d summarize performance comparisons in five criteria (degree of light penetration, rate of energy absorbance, level of uniformity of energy absorbance into a tumor volume, the time required for delivery of targeted light energy, and range of temperature variation) and show distributions of light absorption into a tumor volume for Hypericin and Foscan, respectively”.

Reviewer #2 (Remarks to the Author):

Comments:

In this manuscript, the authors presented AI-informed low-power wireless telemetry with an integrated thermal/light simulation platform for photodynamic therapy (PDT). The AI-assisted wireless telemetry enables adequate illumination of tumors through high-throughput (<20 mice) and multi-wavelength operation. Furthermore, Hypericin and Foscan mediated PDT demonstrates the significant suppression of tumor growth.

The AI-assisted wireless telemetry combined with a supercapacitor-enabled switching mechanism allows for continuous and efficient delivery of multichannel light to mice in a cage and to multiple cages, potentially enabling more efficient and high throughput PDT study in animals. This highly multidisciplinary work is novel and can be of great interest to others in the community. However, some major revisions are needed to clarify and strengthen some of the conclusions.

1. (Major) The motivation/introduction part is somewhat confusing and should be clarified. For example, in the abstract, the authors mentioned the challenge is "Existing approaches... suffer from poor spatiotemporal resolution due to inabilities to minimize oxygen depletion in a tumor." It is not clear why poor spatiotemporal resolution is caused by oxygen depletion – they seem to be two parallel goals. In addition, in the first paragraph, it was mentioned that the current PDT reached a plateau due to poor spatiotemporal resolution. In the second paragraph, it was pointed out that recent advances in wireless technologies improved the spatiotemporal resolution, but the challenges are compensation for light loss versus tissue heat/damage by high intensity light and UHF EM waves. It is somewhat confusing what the proposed technology is trying to achieve – improve the spatiotemporal resolution, minimize oxygen depletion, or reduce tissue heat/damage by using multichannel light sources?

Response: The reviewer’s point is well taken. The technical highlight of this study is an implantable wireless device equipped with a dual-channel function that can be induced by RF power, which significantly improves levels of tissue penetration and PS activation and increases

the throughput of PDT experiments. Key limitations in PDT experiments involve 1) low level of light penetration & PS activation and 2) low throughput. First, existing PDT hardware delivers light to tumors via a cable tethered to a light source and such physical tethering would complicate access to targeted tumor cells. In addition, wired or wireless PDT approaches use single light source (wavelength) to activate a PS and would activate a PS at a modes rate if a PS has two absorption peaks. Together, low levels of light penetration and PS activation would significantly degrade PDT efficacy. Second, current PDT methods are low-yield, and cannot be multiplexed. This is a major constraint that prohibits the high-throughput utilization of wireless methods in most laboratory settings. In this paper, we proposed a novel research platform that integrates Monte Carlo analysis and DeepLabCut algorithm into an implantable wireless telemetry system. Here, the thermal/ light Monte Carlo simulator yields a bespoke PDT regimen such as choice of wavelengths, determination of the number of light sources, and its placement onto an implantable device to be delivered. The DeepLabCut algorithm allows for real-time instantaneous pose estimation of the freely moving animals in a cage (up to 4 cages) to ensure reliable activation of the implanted devices. Collectively, the DeepLabCut-assisted low-power wireless telemetry system uses the parameters from the MC simulator to enable the most effective PDT. This would facilitate efficient use of resources.

2. (Major) Following the previous comment, if one of the major advantages of this technology is to reduce oxygen depletion, some evidence showing that this new method could reduce oxygen depletion in tumors should be provided to support the claim. If the focus is on the capability of delivery light at multiple wavelengths continuously at a low power and the activation of mice in multiple cages, the abstract and the introduction should be modified accordingly.

Response: We agree the abstract and introduction should be revised accordingly. To be clear, we modified the Abstract and 1st and 2nd paragraphs in Introduction section: "Advances in wireless technologies enable remote delivery of light to tumors, but suffer from key limitations, including low levels of tissue penetration and PS activation." & "Together, they establish a range of guidelines for effective PDT regimen design. *In vivo* Hypericin and Foscan mediated PDT, using cancer xenograft models, demonstrated substantial suppression of tumor growth, warranting further investigation in research and/or clinical settings." in Abstract, and "Existing hardware PDT, in particular the light source, suffers from key limitations in their ability to: 1) deliver light to deep tumor cells (e.g., physical tethers to light sources would complicate access to targeted tumor cell), 2) control light sources and/or wavelengths in a programmed manner (e.g., light sources with single wavelength would activate a PS at a modest rate if a PS has two absorption peaks), and 3) be multiplexed so that multiple animals can be studied in parallel.", in 2nd paragraph of Introduction; "Foscan has two peaks at 406 nm and 652 nm in absorption spectra according to different absorption coefficient, respectively.¹⁷. Thus, 78% of light absorbance at 406 nm, which is not normally used for PDT, does not contribute to the generation of oxygen-free radicals and tumor killing. One way to compensate for this is to increase the light intensity, however, its benefit is marginal. Furthermore, increases in light intensity result in heat generation, which can damage normal healthy tissue¹⁸⁻²⁰. Another limitation is the absence of high-throughput pipelines for the analysis of PDT outcomes²¹⁻²³. Most existing wireless approaches utilize an RF power generator for each animal cage. Multiple RF power generators can be used, but they must be operated at least 1m apart to avoid electromagnetic interference²⁴. This is a major constraint that prohibits the high-throughput utilization of wireless methods in most laboratory settings." in 3rd paragraph of Introduction.

3. (Major) Measurement of heat dissipation using IR camera (Supplementary Figure 2): in this figure, the devices are kept inside a bag filled with saline solution, and the IR camera is measuring the temperature from the top. It seems that the camera would be measuring the average temperature of the solution or the bag which may not be the temperature right at or next to the device.

Response: Thank you for your comments about direct measurement of temperature of devices.

A saline solution bag mimics an animal or biological tissues where a device is mounted and justifies the use of it in experiments. IR camera also offers enough spatial resolution (sub mm) to identify a heat source. In addition, we already measured the device directly without solution bag for checking the temperature changes of device itself; this condition represented in Supplementary Figure 2a and the result also can be found in Supplementary Figure 2c. All variation of temperature were below 0.6 °C.

4. (Major) The in vivo experiments compared HYP(+)/LED(+) conditions with controls. However, the multichannel (multiple wavelength) work has only been demonstrated in vitro. Since the multichannel capability is a major contribution of this work, an in vivo demonstration of the multichannel/wavelength activation would be very helpful. Is it realistic to perform in vivo demonstration of the multichannel activation? How might the movement of the implant affect the results?

Response: We agree with reviewer's comment. We conducted animal experiments of each three group with Foscan: 1) red LEDs only, 2) purple LEDs only, and 3) combined dual red/purple colors in a device. As we expected, the tumor's growth was dramatically suppressed by dual color device (Figure 5g). We added the following sentence in the last paragraph of Result 4th subheading section: "To further highlight the advantage of dual-wavelength LED PDT, *in vivo* evaluation was conducted in murine models of HT29 tumor xenografts. After 5 days of continuous PDT treatment, a 0.89-fold (89 %) decrease, a 0.51-fold (51 %) decrease, a 0.64-fold (64%) decrease, and 1.94-fold (194 %) increase in tumor volumes (Fig. 5g)". We have included data in Figure 5. To comment on the movement of the implant, we observed that the implanted devices remained stable and tightly affixed adjacent to the tumor xenografts under the skin following surgical implantation. No implant movement was observed.

5. (Minor) On page 5, the authors use the five criteria (1-5) to choose the optimal condition for PDT. Some sentence describing how these 5 criteria relate to the optimum condition should be added. In addition, "Results performed under different conditions (wavelength) revealed that uniform light delivery and minimum heat dissipation for Hypericin are reached with..." Uniform light delivery and minimum heat dissipation only refers to criteria 3 and 5. A more comprehensive description may also include amount/rate of energy absorption, etc.

Response: We agree with reviewer's comment. We added more details about the criteria and revised followings in Result 2nd subheading section: "Figs. 2c and d summarize performance comparisons in five criteria (degree of light penetration, rate of energy absorbance, level of uniformity of energy absorbance into a tumor volume, the time required for delivery of targeted light energy, and range of temperature variation) and show distributions of light absorption into a tumor volume for Hypericin and Foscan, respectively. Here, we set and calculate these variables from simulation in order to obtain the detailed information on the critical variables for deriving the results: 1) the capability to pass through the type of different medium, such as epidermis, dermis, and tumors, by each light source, 2) the energy absorbance ratio in the tumor according to the light spectra, and 3) the time to reach threshold energy significant for tumor growth suppression. The results, performed under different conditions (wavelength),

revealed that the optimum condition for Hypericin to be most effective in PDT is reached with light sources (wavelength of 590 nm) and 25% of duty-cycle operation while that for Foscan is met with a combination of 406 and 652 nm wavelengths and 25% duty cycle operation. de For Foscan the optimal conditions were with a combination of 406 and 652 nm wavelengths and 25% duty cycle operation. ”.

6. (Minor) In figure 2b, what is the distance between the LED and the tumor? Is it chosen based on the experimental conditions during the *in vitro* testing (such as the condition in Fig.5g)?

Response: Although not mentioned in the Figure 2b, we set the distance between the LED and the tumor to be 50 μm in the simulation. This distance corresponds to the thickness of packaging the implantable device with PDMS, a biocompatible material. In actual *in vitro* and *in vivo* experiments, since the device and the tumor are in contact, the light source and the tumor are kept the distance by the thickness of the packaging material, PDMS. So, we labeled the distance in Figure 2b:

7. (Minor) On page 10, "...power budgets as low as tens of μA ." Why does the power have a unit of μA ? Can the authors estimate the power consumption with the reed switch compared with the power consumption simply using multiple light sources?

Response: As the reviewer mentioned, μA is not a unit of power. We changed the term 'power budgets' to 'current consumption' and added the following sentences to Discussion: "In addition, we designed a low-power circuit to control these multichannel light sources: an actuation mechanism triggered by a reed switch enabled efficient activation of a photosensitizer with current consumption as low as tens of μA . Existing multichannel devices are not suitable for wireless power systems because they require a stable rectified current supply of at least several tens of mA since they contain additional IC chips for multichannel control."

Reviewer #3 (Remarks to the Author):

Comments:

I am not a domain expert in the main subject matter of this paper. I am, however, an expert in animal pose estimation. Thus, I will only comment on these aspects. The combination of antenna tracking and deeplabcut is a nice contribution (and from a non-expert view, the paper is quite exciting). However, I have questions about the AI method being used.

It appears the authors are using a beta release of DeepLabCut code that has not been published (or no preprint yet). Therefore, performance is quite unclear for this method. There are also not sufficient details in the method to understand what the authors used exactly. In the main text they mention using neural networks for pose estimation in multiple animals, and then in the methods they say deeplabcut 2.2b7-- looking at this code it's clear there are many options that are not clearly described in the methods. Moreover, the authors mention bipartite matching but the methods don't detail this fully. It seems unusual, to me, that the authors here are using unpublished code without proper benchmarking and performance metrics. I give more concrete examples below.

"Modification to DLC model

We utilized, custom-trained, and modified the DLC Python package (Ver. 2.2b7). Specifically, we used the custom-trained DLC model to estimate the locations of the body parts such as snouts and tails of the mice within an image (i.e., video frame). Note that the original DLC python package does not support a real-time processing feature, instead it only runs on video files. Hence, we directly modified the Python package in such a way that it can infer the locations of the body parts of the mice and estimate the optimal coil antenna through the functional modules, illustrated in Fig. 3a, in a real-time manner. We conducted all experiments including training the DLC on a GPU workstation (Lambda workstation with Intel Core i9-9960X, 128 GB RAM, and two GEFORCE RTX 2080 Ti graphics cards)."

1. (Major) It appears the authors are using unpublished DeepLabCut code: 2.2beta7 is a beta release. It would seem appropriate to ask those authors how to appropriately cite this version? Specifically, Figure 3A 2-4 seems to be closely related to the unpublished multi-animal DeepLabCut contribution. DLC has 14 networks available, multi-data augmentation steps, new steps introduced in 2.2.beta, etc. The authors should give sufficient details to be clear to the reader what they used.

Response: Thank you for your comment regarding the citation of DeepLabCut.

First of all, we would like to emphasize that Figure 3a.(3)-(6) illustrates the contribution we have made to this work, which can run on the original (previous) version of DeepLabCut as well. Thus, I believe it will suffice to cite the original DLC manuscript. While we have developed the proposed software based on the latest beta version (2.2b7) of the DeepLabCut package, as it is more stable than the original version, we do not utilize or rely on its new features that were not included in the original DLC that we cited in our manuscript.

The primary objective of the software that we developed was to efficiently identify the optimal power transmission antenna based on the "instantaneous poses" of the multiple mice in a real-time manner. In that regard, we believe it may be worthwhile to briefly discuss the characteristics of the new feature in the latest DeepLabCut (referred to as MT) for tracking multiple animals. MT is indeed a multiple animal instance "tracking" post-processing algorithm that requires manual interventions from a user, which cannot be easily converted to a real-time processing setting – which is our main focus. In fact, the MT feature works only for offline processing of recorded video files. Besides, for the purpose of optimal antenna control, our method only requires real-time identification of the "instantaneous poses" of multiple mice without the need for continuously tracking them.

To this aim, we trained the DeepLabCut model, which was already implemented in the original (previous) DeepLabCut package cited in our manuscript and used the model only to detect the locations of the body parts that are not yet associated with the instance label (for individual animal). Our proposed real-time post-processing module, illustrated in Figure 3a.(3)-(6), "intercepts" the raw output from the output layer of the DLC model, based on which it predicts the optimal power transmission coil antenna. To be more specific, our algorithm

computes the matching score based on the weighted Euclidean distance between all possible pairs of different body parts (i.e., all possible head-tail pairs). Then, it finds the optimal one-to-one mapping through the maximum weighted bipartite matching (MWBM) to identify the instantaneous poses (directions) of the multiple mice. The optimal transmission antenna for each animal is identified according to the direction of each optoelectronic device, and the globally optimal power transmission coil antenna is determined based on the majority rule.

To clearly describe our contribution and avoid any potential confusion, we updated the terminologies in the manuscript accordingly (motion tracking → **instantaneous pose estimation**). In addition, we changed the title of the corresponding section “AI-enabled real-time motion tracking of multiple animals” to “**Real-time identification of the optimal coil antenna based on the instantaneous pose estimation of multiple animals**” and updated the content as follows:

Real-time identification of the optimal coil antenna based on the instantaneous pose estimation of multiple animals

We developed software that identifies the optimal power transmission coil antenna in a real-time manner in the sense that the number of optoelectronic devices receiving wireless power is maximized. The proposed software is built on the strength of the state-of-the-art deep learning model, DLC²⁵, combined with the proposed real-time post-processing module based on the optimal graph matching algorithm, maximum weighted bipartite matching (MWBM)²⁶. To be more specific, we directly customized the DLC Python package such that the custom-trained DLC model runs with a real-time video stream. The proposed real-time post-processing module intercepts the raw estimated locations of the body parts (heads and tails) of the mice, where the estimated locations are not associated with the instance label, from the output layer, and estimates instantaneous poses (directions) of mice. Based on the estimated poses, it identifies the optimal coil antenna.

Fig. 3a illustrates the procedures of the proposed software. As shown in Fig. 3a-(1), we assume that there are five freely behaving mice in which an optoelectronic device has been implanted. For the wireless power transmission, four power transmission coil antennas, only one of which activates at a time, surround the cage while forming an X-shape to all sides. A webcam on the top of the cage sends a video stream to the custom-trained DLC at the rate of 25 fps. As a frame arrives, the trained DLC model detects the locations of the snouts or tails (Fig. 3a-(2)). Note that the raw output (i.e., the locations of the snouts or tails) from the output layer of DLC are not associated with the instance information. The proposed software intercepts the raw output, filters out the detected body parts of which the confidence score is less than 0.6, and passes it to the proposed real-time post-processing module. Next, the proposed post-processing module computes the matching score between all possible pairs consisting of different body parts (i.e., head-tail pair) (Fig. 3a-(3)). Based on the matching scores, MWBM finds the optimal one-to-one pairs in such a way that the sum of the matching scores of the pairs in the optimal matching set is maximized (Fig. 3a-(4)). The optimal pairs directly correspond to the instantaneous poses (direction) of the mice in the cage (Fig. 3a-(5)). In turn, for each optimal pair or, equivalently, the direction of each mouse, the optimal coil antenna which is expected to achieve maximum power efficiency to each device is selected (Supplementary Fig. 5). Finally, the global optimal power transmission coil antenna is selected and activated according to the rule of the majority (Fig. 3a-(6)). The software repeats these whole procedures throughout the experiment.

For assessment of the proposed software, we use a metric, defined as the percentage of correct predictions for the data tested²⁷. Fig. 3b shows antenna selection accuracy for three different antenna settings: 1) two pairs of X-shaped antenna coils, one pair of X-shaped coil aligning with 2) the x- or 3) y-axis. Results revealed that the proposed software guarantees the accuracy of 80 % or above in every setting that we tested (Fig. 3b). Figs. 3c and d show

statistics of the number of frames for two representative cases; how long a selected antenna remains activated (Fig. 3c) and how many frames (or long interval times) it takes between activation of an antenna and reactivation of itself after the first deactivation (Fig. 3d). Likely, some occupants, not all of them, in a cage may not receive enough power for activation of a photosensitizer due to a decision by the software (e.g., when two mice or vectors along the length of their body are aligned with the x-axis and corresponding vectors for the rest three mice are on the y-axis, the software chooses an antenna coil that offers better wireless coverage in the y-direction). Experimental results revealed that discharges of power stored in an embedded supercapacitor can last longer than the time intervals shown in Supplementary Figs. 7 and 8. This suggests that the proposed software paired with the use of a supercapacitor ensures robust activation of devices in a cage. Detailed information on evaluations of the DeepLabCut software is found in Method section.

2. (Major) There are also several papers that present real-time options for DeepLabCut; can the authors comment on the differences here from Forys et al, Kane et al, etc? (Technically, DLC could always take in frames, whether in a video or frame format-- it is a frame-by-frame pose estimation algorithm that always allowed for batch size 1). Note that in Kane et al, 2020 there is also a novel predictive model to reduce delays in low-latency tracking.

Response: Thank you for your insightful comment on the recent works relevant to DLC. Conceptually, the recent works by Forys et al. and Kane et al. and our proposed algorithm bear some high-level similarities. However, our proposed algorithm has distinctive characteristics that clearly differentiate it from the works you've mentioned. We delineate the important differences in what follows.

The work by Forys et al. is relevant to our proposed algorithm in the sense that they both aim to control the experimental platform in a real-time manner based on the behavior of the animal subjects by modifying the DLC package. Forys et al. developed software that tracks the movement of specific body parts (i.e., paws) of a mouse in a real-time manner and activates the indicative device if the degree of activity exceeds a certain threshold with very low latency. However, despite the outward similarity, there is a critical difference between the work by Forys et al. and our proposed algorithm. For example, the approach by Forys et al. has been restricted to tracking body parts of a single mouse whose head is fixed. On the other hand, our proposed algorithm estimates the instantaneous poses of multiple mice, based on which we identify the optimal power transmission coil antenna that effectively delivers power to the wireless optoelectronic devices. This is clearly beyond the capability of the original DLC as well as the software developed by Forys et al.

Kane et al. developed a lightweight python package called DeepLabCut-live! (DLC-live) based on the DLC, mainly aiming at extending the applicability of the DLC for diverse experimental studies that typically involve monitoring the behaviors of an animal and intervening in the experiment in a real-time manner. However, to date, DLC-live has been mainly validated for tracking a single animal. Furthermore, considering that the multiple animal tracking feature introduced in the latest DLC requires frequent user intervention, it is not straightforward to combine DLC-live with the multiple animal tracking feature in the latest DLC. Also recall that we are only interested in identifying the optimal power transmission coil antenna based on the "instantaneous poses" of the multiple mice, rather than real-time tracking of their temporal trajectories. Our proposed approach effectively accomplishes this primary objective in a real-time setting, which is not possible by the original DLC or DLC-live.

3. (Major) The authors should be sure to cite the open source code they use, such as Matplotlib, numpy, and importantly the DLC Python package of the code (Nath et al, 2019).

Response: Thank you for the valuable comment. DLC software provides the Conda environmental file that contains the information of all packages utilized by DLC. In addition to

the packages, we utilized the NetworkX package in order to construct a body parts network and to predict the instantaneous poses by predicting the snout-tail pairs using the maximum weighted bipartite matching (MWBM) scheme. The resulting python package information is summarized in the table below. We included the following table in the supplementary information and added the following comments in Method section: "We utilized, custom-trained, and modified the DLC Python package (Ver. 2.2b7); the summary of Python package information is in Supplementary Table 1."

Supplementary Table 1. Summary of resulting python package information

Package name	Version	Package name	Version	Package name	Version	Package name	Version
absi-py	0.9.0	msgpack	1.0.0	imgaug	0.4.0	qt	5.9.7
argon2-cffi	20.1.0	msgpack-numpy	0.4.6.1	importlib-metadata	1.7.0	qtconsole	4.7.5
astor	0.8.1	nb_conda	2.2.1	importlib_metadata	1.7.0	qtpy	1.9.0
astroid	2.4.2	nb_conda_kernels	2.2.3	intel-openmp	2020.0.133	readline	8
attrs	19.3.0	nbconvert	5.6.1	ipykernel	5.3.4	requests	2.24.0
backcall	0.2.0	nbformat	5.0.7	ipython	7.17.0	ruamel-yaml	0.16.10
bayesian-optimization	1.2.0	ncurses	6.2	ipython_genutils	0.2.0	ruamel-yaml-clib	0.2.0
blas	1	networkx	2.4	ipywidgets	7.5.1	scikit-image	0.17.2
bleach	3.1.5	notebook	6.1.1	isort	5.4.2	scikit-learn	0.23.2
c-ares	1.15.0	numba	0.51.0	jdcalf	1.4.1	scipy	1.5.2
ca-certificates	2020.7.22	numexpr	2.7.1	jedi	0.17.2	send2trash	1.5.0
cairo	1.14.12	numpy	1.16.4	jinja2	2.11.2	setuptools	49.6.0
certifi	2020.6.20	numpy-base	1.19.1	joblib	0.16.0	shapely	1.7.0
cffi	1.14.1	opencv-python	4.1.2.30	jpeg	9b	sip	4.19.8
chardet	3.0.4	openpyxl	3.0.5	jsonschema	3.2.0	six	1.15.0
click	7.1.2	openssl	1.1.1g	jupyter	1.0.0	sqlite	3.33.0
cuda-toolkit	10.0.130	packaging	20.4	jupyter_client	6.1.6	statsmodels	0.11.1
cuda-nn	7.6.5	pandas	1.1.1	jupyter_console	6.1.0	tables	3.6.1
cuDNN	10.0.130	pandoc	2.10.1	jupyter_core	4.6.3	tabulate	0.8.7
cycler	0.10.0	pandocfilters	1.4.2	keras-applications	1.0.8	tensorboard	1.13.1
cython	0.29.21	pango	1.45.3	keras-preprocessing	1.1.0	tensorflow	1.13.1
dbus	1.13.16	parso	0.7.0	kiwisolver	1.2.0	tensorflow-base	1.13.1
decorator	4.4.2	patsy	0.5.1	lazy-object-proxy	1.4.3	tensorflow-estimator	1.13.0
deeplabcut	2.2b7	pcre	8.44	ld_impl_linux-64	2.33.1	tensorflow-gpu	1.13.1
defusedxml	0.6.0	pexpect	4.8.0	libedit	3.1.20191231	tensorpack	0.10.1
easydict	1.9	pickleshare	0.7.5	libffi	3.3	termcolor	1.1.0
entrypoints	0.3	pillow	7.2.0	libgcc-ng	9.1.0	terminado	0.8.3
et-xmlfile	1.0.1	pip	20.2.2	libgfortran-ng	7.3.0	testpath	0.4.4
expat	2.2.9	pixman	0.40.0	libglu	9.0.0	threadpoolctl	2.1.0
filterpy	1.4.5	proglog	0.1.9	libpng	1.6.37	tifffile	2020.8.13
fontconfig	2.13.0	prometheus_client	0.8.0	libprotobuf	3.13.0	tk	8.6.10
freetype	2.10.2	prompt-toolkit	3.0.5	libsodium	1.0.18	toml	0.10.1
fribidi	1.0.10	prompt_toolkit	3.0.5	libstdcxx-ng	9.1.0	tornado	6.0.4
gast	0.4.0	protobuf	3.13.0	libuuid	1.0.3	tqdm	4.48.2
geos	3.8.0	psutil	5.7.2	libxcb	1.14	traitlets	4.3.3
gettext	0.19.8.1	ptyprocess	0.6.0	libxml2	2.9.10	typed-ast	1.4.1
glib	2.65.0	pycparser	2.2	lvm-lite	0.34.0	urllib3	1.25.10
graphite2	1.3.14	pygments	2.6.1	markdown	3.2.2	wcwidth	0.2.5
grpcio	1.31.0	pylint	2.6.0	markupsafe	1.1.1	webencodings	0.5.1
gst-plugins-base	1.14.0	pyparsing	2.4.7	matplotlib	3.0.3	werkzeug	1.0.1
gstreamer	1.14.0	pyqt	5.9.2	mccabe	0.6.1	wheel	0.34.2
h5py	2.10.0	pyrsistent	0.16.0	mistune	0.8.4	widetsnbextension	3.5.1
harfbuzz	2.4.0	python	3.7.7	mkl	2020.1	wrapt	1.11.2
hdf5	1.10.6	python-dateutil	2.8.1	mkl-service	2.3.0	wxpython	4.0.4
icu	58.2	pytz	2020.1	mkl_fft	1.1.0	xz	5.2.5
idna	2.1	pywavelets	1.1.1	mkl_random	1.1.1	zeromq	4.3.2
imageio	2.9.0	pyyaml	5.3.1	mock	4.0.2	zipp	3.1.0
imageio-ffmpeg	0.4.2	pyzmq	19.0.1	moviepy	1.0.1	zlib	1.2.11

4. (Minor) “The utilization of an advanced AI algorithm for automated video analysis allows for real time tracking of the freely moving animals in a cage to ensure robust activation of animals (implantable devices) in cages” -- the authors might consider stating what they used, vs. the flashy and non-informative “AI”; simply stating they used DeepLabCut is sufficient.

Response: We appreciate your thoughtful suggestion. In the revised manuscript, we changed “AI” to “DeepLabCut” according to your comment: for example in title, “DeepLabCut-based, implantable, multichannel wireless telemetry for photodynamic therapy” in abstract, “Here, we introduce DeepLabCut (DLC)-informed low-power wireless telemetry with an integrated thermal/light simulation platform that overcomes the above constraints. The simulator produces an optimized combination of wavelengths and light sources, and DLC-assisted wireless telemetry uses the parameters from the simulator to enable adequate illumination of tumors through high-throughput (<20 mice) and multi-wavelength operation.” In introduction, “The utilization of an advanced DLC algorithm for automated video analysis allows for real-time instantaneous pose estimation of the freely moving animals in a cage to ensure robust activation of animals (implantable devices) in cages.” in method, “The proposed DLC algorithm yields the following information for each frame”.

5. (Minor) Figure 3. (a) Illustration of step-by-step procedures for the proposed artificial intelligence algorithm. -- this is not a newly "proposed" software (this is DeepLabCut), so please consider revisiting the figure caption and also be sure to include a TM to the DLC image you have in Figure 3.

Response: We appreciate your suggestion regarding Figure 3. As we answered above (esp., see answers to the first and second comments), our proposed software consists of two core modules: (1) a custom trained DLC and (2) a post-processing module that takes the raw output from the output layer of the DLC for instantaneous pose estimation of multiple animals and the optimal antenna prediction. It should be noted that that there is no instance information in the raw output from the DLC that connects the predicted body parts to individual mice. Our contribution is to find the optimal pairs (i.e., snout-tail pairs) via maximum weighted bipartite matching (MWBM), thereby estimating the directions of up to five mice. Based on the directions of the mice identified by our algorithm, we predict the optimal power transmission antenna that maximizes the power transmission efficiency. For this reason, we prefer to maintain the "proposed" software in Figure 3 as is, since this is an important new capability that cannot be simply achieved by the DLC without significant development and optimization. Nevertheless, we certainly agree with your suggestion that it would be important to clearly show the role of DLC in our proposed software. Following your suggestion, we included TM in the DLC image and updated Figure 3a.(2) to clearly specify our contribution.

Reviewers' Comments:

Reviewer #1:

Remarks to the Author:
satisfactory revision

Reviewer #2:

Remarks to the Author:

The authors have addressed a lot of my concerns, and the quality of the manuscript has been improved. However, in the newly added in vivo experiment (Fig.5g), $n=2$ is not sufficient to draw a conclusion. A minimum of $n \geq 3$ should be used in these experiments.

Reviewer #3:

Remarks to the Author:

I thank the authors for their revisions, but I still find the methods unacceptably lacking details and missing citations. This must be fixed prior to any publication.

Authors comments:

"First of all, we would like to emphasize that Figure 3a.(3)-(6) illustrates the contribution we have made to this work, which can run on the original (previous) version of DeepLabCut as well."

My response:

You use code that is absolutely not the original DLC and you directly use and slightly modify the unpublished multianimal code (I read and compared the code e.g. this line in their code specifically states it is DLC 2.X: https://github.com/parkgroup-tamu/AI-enabled-implantable-multichannel-wireless-telemetry-for-photodynamic-therapy/blob/8a0a6f78e6d7f221718530664aa76dee9669dd84/DeepLabCut_Modified/predict_multi_animal.py#L7); as someone who is very familiar with the DLC code base. I do not know why these authors do not feel it's necessary to provide proper credit to the authors of DeepLabCut. <https://github.com/DeepLabCut/DeepLabCut#references> which states:

"If you use this code or data we kindly ask that you please cite Mathis et al, 2018 and, if you use the Python package (DeepLabCut2.x) please also cite Nath, Mathis et al, 2019. If you utilize the MobileNetV2s or EfficientNets please cite Mathis, Biasi et al. 2021. If you use multi-animal in beta mode, please contact us; if you use the 2.2rc1+, please cite Lauer et al. 2021."

In short, you clearly use 2.x+ code, yet do not cite Nath et al 2019? You clearly used their beta code, and don't cite their multi-animal preprint (Lauer et al 2021), which came out before this revision/submission? I would also suggest citing the DLC-Live version and other real-time deeplabcut-based packages such as DeepLabStream, since your method has synergies with this (and clearly is directly inspired by the code in <https://github.com/DeepLabCut/DeepLabCut-live>, and since you used the beta code of 2.2 DLC and it's multi-animal aspects, you should cite Lauer et al 2021.

There is no limitation from Nature Communications that prevents this.

Moreover, you state you "we directly customized the DLC Python package;" if you do this, their GNU Lesser General Public License v3.0 license states you must make your source code fully open source and provide the original license. Note, the link is again not functional in the manuscript, but searching for the code I find it violates the terms of the original DLC license:

https://github.com/parkgroup-tamu/AI-enabled-implantable-multichannel-wireless-telemetry-for-photodynamic-therapy/tree/main/DeepLabCut_Modified  this does not include a fully copy of the license.

The docs here: https://github.com/parkgroup-tamu/AI-enabled-implantable-multichannel-wireless-telemetry-for-photodynamic-therapy/tree/main/DeepLabCut_Modified also state that this is based on DLC 2.2b7, again clearly contradicting the response to the reviewers.

Lastly, I find it usual they changed the title of the paper to be "DeepLabCut..."-- the authors are not contributors to the DeepLabCut on GitHub (I checked), nor do they hold TM rights to the name; I would suggest changing this back to avoid any issues.

Missing details example such that the work is not reproducible:

"Modification to DLC model

372 We utilized, custom-trained, and modified the DLC Python package (Ver. 2.2b7); the summary
373 of Python package information is in Supplementary Table 1. Specifically, we used the
custom374 trained DLC model to estimate the locations of the body parts such as snouts and tails
of the
375 mice within an image (i.e., video frame). Note that the original DLC python package does not
376 support a real-time processing feature, instead it only runs on video files. Hence, we directly
377 modified the Python package in such a way that it can infer the locations of the body parts of
the
378 mice and estimate the optimal coil antenna through the functional modules, illustrated in Fig.
3a,
379 in a real-time manner. We conducted all experiments including training the DLC on a GPU
380 workstation (Lambda workstation with Intel Core i9-9960X, 128 GB RAM, and two GEFORCE
381 RTX 2080 Ti graphics cards)."

There are 14 backbone architectures in DeepLabCut, 3 data augmentation methods, and many hyperparameters. What did they use? How many images are labeled? How long was the model trained? How did they select the snapshot they used? I would, again, highly suggest you take the recommendation to be specific. Here are guidelines for reproducible reporting of DeepLabCut-based models: <https://deeplabcut.github.io/DeepLabCut/docs/recipes/DLCMethods.html>

Reviewer #2 (Remarks to the Author):

1. The authors have addressed a lot of my concerns, and the quality of the manuscript has been improved. However, in the newly added in vivo experiment (Fig.5g), $n=2$ is not sufficient to draw a conclusion. A minimum of $n \geq 3$ should be used in these experiments.

Response: It was a typo and we fixed it ($n=5$) on the manuscript.

Reviewer #3 (Remarks to the Author):

I thank the authors for their revisions, but I still find the methods unacceptably lacking details and missing citations. This must be fixed prior to any publication.

1. You use code that is absolutely not the original DLC and you directly use and slightly modify the unpublished multianimal code (I read and compared the code e.g. this line in their code specifically states it is DLC 2.X: https://github.com/parkgroup-tamu/AI-enabled-implantable-multichannel-wireless-telemetry-for-photodynamic-therapy/blob/8a0a6f78e6d7f221718530664aa76dee9669dd84/DeepLabCut_Modified/predict_multianimal.py#L7); as someone who is very familiar with the DLC code base. I do not know why these authors do not feel it's necessary to provide proper credit to the authors of DeepLabCut. <https://github.com/DeepLabCut/DeepLabCut#references> which states:

"If you use this code or data we kindly ask that you please cite Mathis et al, 2018 and, if you use the Python package (DeepLabCut2.x) please also cite Nath, Mathis et al, 2019. If you utilize the MobileNetV2s or EfficientNets please cite Mathis, Biasi et al. 2021. If you use multi-animal in beta mode, please contact us; if you use the 2.2rc1+, please cite Lauer et al. 2021."

In short, you clearly use 2.x+ code, yet do not cite Nath et al 2019? You clearly used their beta code, and don't cite their multi-animal preprint (Lauer et al 2021), which came out before this revision/submission? I would also suggest citing the DLC-Live version and other real-time deeplabcut-based packages such as DeepLabStream, since your method has synergies with this (and clearly is directly inspired by the code in <https://github.com/DeepLabCut/DeepLabCut-live>, and since you used the beta code of 2.2 DLC and it's multi-animal aspects, you should cite Lauer et al 2021.

There is no limitation from Nature Communications that prevents this.

Response: Thank you very much for your kind suggestions and the detailed information. We absolutely agree with you that it is critically important to give due credit to the DLC authors that they deserve and acknowledge their work. We regret we were not more careful when referencing the DLC-related publications in our original manuscript. Following your suggestions, in the revised manuscript, we cited all the relevant references that you mentioned in your comments. Specifically, we cited the manuscripts by Mathis et al [25], Nath, T. et al. 2019 [26], and Lauer, J. et al 2021 [27] in our manuscript. Regarding converting the DLC for real-time processing, we would like to emphasize that we already converted and utilized DLC for real-time processing in our previous study [46] (presented in Nov. 2020 @ 2020 54th Asilomar Conference on Signals, Systems, and Computers, <https://ieeexplore.ieee.org/abstract/document/9443501>) before the DLC-Live (Dec. 2020) and DeepLabStream (2021). However, since our main contribution in this work is less relevant to

converting the source code for real-time processing, we believe it will be worth introducing the recent efforts to the readers. In that regard, we cited DLC-Live [47] and DeepLabStream [48] in our manuscript as follows:

DeepLabCut model training

We trained the DLC model provided in the DLC Python package (version 2.2b7^{25,26,27}) such that the trained model is capable of locating snouts and tails of up to five mice within an image (*i.e.*, video frame). To this aim, we extracted 30 frames from a video file recorded for 10 minutes 15 seconds (15,375 frames in total) at the framerate of 25 frames per second via a K-means clustering algorithm to collect representative training frames. We used a ResNet-50^{43,44} neural network with default parameters. For example, we optimized the network via ADAM⁴⁵ with 200,000 iterations and a gradually decreasing learning rate. As a result, the trained model achieved a validation loss of 0.0013. For all details, see config.yaml on the GitHub repository (https://github.com/parkgroup-tamu/AI-enabled-implantable-multichannel-wireless-telemetry-for-photodynamic-therapy/tree/main/DeepLabCut_Modified). We conducted all experiments including training the DLC on Lambda workstation with Intel Core i9-9960X, 128 GB RAM, and two GEFORCE RTX 2080 Ti graphics cards. All python packages used in this study are summarized in Supplementary Table 1.

Modification of DeepLabCut Python package for identifying the optimal coil antenna

In order to identify the optimal transmission coil antenna that maximizes instantaneous system power transmission efficiency online, we directly modified the DLC Python package (Ver. 2.2b7)^{25,26,27}. Specifically, as the original DLC python package does not support a real-time processing feature, for each frame, we intercepted the raw estimation results (*i.e.*, the locations of snouts and tails without instance information) from the trained DLC model and used them as input to the real-time post-processing module we developed as illustrated in Fig. 3a, which was inspired by our previous study⁴⁶. It is worth noting that there have been several attempts to add a real-time processing feature to DLC and platform it^{47,48}. Although the software developed in this study was not built on these platforms to minimize potential unnecessary overhead, technically and conceptually, the modified DLC Python package has in common with these works.

As a frame arrives at the trained DLC model (Fig. 3a-(2)), the trained DLC model identifies the snouts and tails with confidence scores. In this study, we assumed that the body part whose score exceeds 0.6 is valid. As shown in Fig. 3a-(3), the proposed real-time post-processing module computes matching score $s(i, j)$ among all possible pairs of snout and tails to find the optimal pairs as follows:

$$s(i, j) = \left(w_{i,j} \sqrt{(x_{T,j} - x_{S,i})^2 + (y_{T,j} - y_{S,i})^2} \right)^{-1}, \quad (1)$$

where $w_{i,j} = \begin{cases} 1, & \text{if } c(i, j) < T_c \\ \infty, & \text{if } c(i, j) \geq T_c \end{cases}$; $x_{T,j}$ and $x_{S,i}$ are the positions of the j th tail and i th snout on the x -axis, respectively; and $y_{T,j}$ and $y_{S,i}$ are the positions of the j th tail and i th snout on the y -axis, respectively. $c(i, j)$ is the number of pixels, whose intensity value is greater than or equal to T_c , on the straight line connecting two points $(x_{S,i}, y_{S,i})$ and $(x_{T,j}, y_{T,j})$. Next, the proposed module finds the optimal one-to-one mapping set \mathcal{M}^* via the MWBM as follows (Fig. 3a-(4)):

$$\mathcal{M}^* = \operatorname{argmax}_{\mathcal{M}} \left(\sum_{(i,j) \in \mathcal{M}} s(i, j) \right). \quad (2)$$

Construction of the optimal one-to-one mapping set \mathcal{M}^* naturally leads to identifying the projection vector $\mathbf{v}_j | (x_j, y_j)$ of the orientation vector \mathbf{v}_j on the xy -plane of the j th mouse. Under

the practically reasonable assumption that a mouse can only change the angle up to 90 degrees in the direction the mouse are looking (i.e., mice do not stand on their hands), the optimal antenna index i_j^* for j th mouse can be found as follows:

$$i_j^* = \operatorname{argmin}_{i \in \{1, 2, 3, 4\}} E_Z[\|v_j(x_j, y_j) - a_i\|], \quad (3)$$

where a_i , $i = 1, 2, 3, 4$, is the antenna vector of the i th coil antenna (See supplementary Fig. 5). Finally, the optimal coil antenna that maximize the instantaneous system power transmission efficiency is determined by the principle of majority vote.

25. Mathis, A. *et al.* DeepLabCut: markerless pose estimation of user-defined body parts with deep learning. *Nat. Neurosci.* **21**, 1281–1289 (2018).
26. Nath, T. *et al.* Using DeepLabCut for 3D markerless pose estimation across species and behaviors. *Nat. Protoc.* **14**, 2152–2176 (2019).
27. Lauer, J. *et al.* Multi-animal pose estimation and tracking with DeepLabCut. *bioRxiv* (2021).42. Insafutdinov, E. *et al.* DeeperCut: A deeper, stronger, and faster multi-person pose estimation model. *In European Conference on Computer Vision*, 34-50 (Springer, 2016).
43. Insafutdinov, E. *et al.* DeeperCut: A deeper, stronger, and faster multi-person pose estimation model. *In European Conference on Computer Vision*, 34-50 (Springer, 2016).
44. He, K., Zhang, X., Ren, S. & Sun, J. Deep residual learning for image recognition. *In Proceedings of the IEEE conference on computer vision and pattern recognition*, 770-778 (2016).
45. Kingma, D. P. & Ba, J. Adam: A method for stochastic optimization. *arXiv preprint arXiv:1412.6980* (2014).
46. Woo, H.-M. *et al.* Machine Learning Enabled Adaptive Wireless Power Transmission System for Neuroscience Study. *2020 54th Asilomar Conference on Signals, Systems, and Computers*, pp. 808-812 (2020).
47. Kane G. A. *et al.* Real-time, low-latency closed-loop feedback using markerless posture tracking. *Elife* **9**, e61909 (2020).
48. Schweihoff, J. F. *et al.* DeepLabStream enables closed-loop behavioral experiments using deep learning-based markerless, real-time posture detection. *Commun Biol* **4**, 130 (2021).

2. Moreover, you state you "we directly customized the DLC Python package;" if you do this, their GNU Lesser General Public License v3.0 license states you must make your source code fully open source and provide the original license. Note, the link is again not functional in the manuscript, but searching for the code I find it violates the terms of the original DLC license: https://github.com/parkgroup-tamu/AI-enabled-implantable-multichannel-wireless-telemetry-for-photodynamic-therapy/tree/main/DeepLabCut_Modified  this does not include a fully copy of the license.

The docs here: https://github.com/parkgroup-tamu/AI-enabled-implantable-multichannel-wireless-telemetry-for-photodynamic-therapy/tree/main/DeepLabCut_Modified

also state that this is based on DLC 2.2b7, again clearly contradicting the response to the reviewers.

Response: We appreciate your careful review and valuable comments regarding the license. In fact, the URL address was correct, but the hyperlink was broken while converting the manuscript to the PDF version. We apologize for the problem you have experienced, and we will ensure that all links are functional when submitting the revised manuscript. Following your suggestion, we have now included the license (GNU LESSER GENERAL PUBLIC LICENSE Version 3) in the repository. We updated the Code availability section as follows:

Code availability

The code and the trained DLC model are available from GitHub (https://github.com/parkgroup-tamu/Al-enabled-implantable-multichannel-wireless-telemetry-for-photodynamic-therapy/tree/main/DeepLabCut_Modified).

3. Lastly, I find it usual they changed the title of the paper to be "DeepLabCut..."-- the authors are not contributors to the DeepLabCut on GitHub (I checked), nor do they hold TM rights to the name; I would suggest changing this back to avoid any issues.

Response: Thank you very much for your suggestion. We have changed the title of the manuscript back to the original title "AI-enabled, implantable, multichannel wireless telemetry for photodynamic therapy."

4. Missing details example such that the work is not reproducible:

"Modification to DLC model

372 We utilized, custom-trained, and modified the DLC Python package (Ver. 2.2b7); the summary
373 of Python package information is in Supplementary Table 1. Specifically, we used the custom374
trained DLC model to estimate the locations of the body parts such as snouts and tails of the
375 mice within an image (i.e., video frame). Note that the original DLC python package does not
376 support a real-time processing feature, instead it only runs on video files. Hence, we directly
377 modified the Python package in such a way that it can infer the locations of the body parts of the
378 mice and estimate the optimal coil antenna through the functional modules, illustrated in Fig. 3a,
379 in a real-time manner. We conducted all experiments including training the DLC on a GPU
380 workstation (Lambda workstation with Intel Core i9-9960X, 128 GB RAM, and two GEFORCE
381 RTX 2080 Ti graphics cards)."

There are 14 backbone architectures in DeepLabCut, 3 data augmentation methods, and many hyperparameters. What did they use? How many images are labeled? How long was the model trained? How did they select the snapshot they used? I would, again, highly suggest you take the recommendation to be specific. Here are guidelines for reproducible reporting of DeepLabCut-based models: <https://deeplabcut.github.io/DeepLabCut/docs/recipes/DLCMethods.html>

Response: We appreciate your valuable comment on the additional information that needs to be provided to ensure reproducibility. Following your suggestions, we removed the "Modification to DLC model" subsection in the Methods section and added two subsections "DeepLabCut model training" and "Modification of DeepLabCut Python package for identifying the optimal coil antenna" to provide the full details to readers. Besides, we explicitly mentioned in the subsection that the trained model and all the hyperparameters are available on our GitHub repository as follows:

DeepLabCut model training

We trained the DLC model provided in the DLC Python package (version 2.2b7^{25,26,27}) such that the trained model is capable of locating snouts and tails of up to five mice within an image (i.e.,

video frame). To this aim, we extracted 30 frames from a video file recorded for 10 minutes 15 seconds (15,375 frames in total) at the framerate of 25 frames per second via a K-means clustering algorithm to collect representative training frames. We used a ResNet-50^{43,44} neural network with default parameters. For example, we optimized the network via ADAM⁴⁵ with 200,000 iterations and a gradually decreasing learning rate. As a result, the trained model achieved a validation loss of 0.0013. For all details, see config.yaml on the GitHub repository (https://github.com/parkgroup-tamu/AI-enabled-implantable-multichannel-wireless-telemetry-for-photodynamic-therapy/tree/main/DeepLabCut_Modified). We conducted all experiments including training the DLC on Lambda workstation with Intel Core i9-9960X, 128 GB RAM, and two GEFORCE RTX 2080 Ti graphics cards. All python packages used in this study are summarized in Supplementary Table 1.

Modification of DeepLabCut Python package for identifying the optimal coil antenna

In order to identify the optimal transmission coil antenna that maximizes instantaneous system power transmission efficiency online, we directly modified the DLC Python package (Ver. 2.2b7)^{25,26,27}. Specifically, as the original DLC python package does not support a real-time processing feature, for each frame, we intercepted the raw estimation results (*i.e.*, the locations of snouts and tails without instance information) from the trained DLC model and used them as input to the real-time post-processing module we developed as illustrated in Fig. 3a, which was inspired by our previous study⁴⁶. It is worth noting that there have been several attempts to add a real-time processing feature to DLC and platform it^{47,48}. Although the software developed in this study was not built on these platforms to minimize potential unnecessary overhead, technically and conceptually, the modified DLC Python package has in common with these works.

As a frame arrives at the trained DLC model (Fig. 3a-(2)), the trained DLC model identifies the snouts and tails with confidence scores. In this study, we assumed that the body part whose score exceeds 0.6 is valid. As shown in Fig. 3a-(3), the proposed real-time post-processing module computes matching score $s(i, j)$ among all possible pairs of snout and tails to find the optimal pairs as follows:

$$s(i, j) = \left(w_{i,j} \sqrt{(x_{T,j} - x_{S,i})^2 + (y_{T,j} - y_{S,i})^2} \right)^{-1}, \quad (1)$$

where $w_{i,j} = \begin{cases} 1, & \text{if } c(i, j) < T_c \\ \infty, & \text{if } c(i, j) \geq T_c \end{cases}$; $x_{T,j}$ and $x_{S,i}$ are the positions of the j th tail and i th snout on the x -axis, respectively; and $y_{T,j}$ and $y_{S,i}$ are the positions of the j th tail and i th snout on the y -axis, respectively. $c(i, j)$ is the number of pixels, whose intensity value is greater than or equal to T_c , on the straight line connecting two points $(x_{S,i}, y_{S,i})$ and $(x_{T,j}, y_{T,j})$. Next, the proposed module finds the optimal one-to-one mapping set \mathcal{M}^* via the MWBM as follows (Fig. 3a-(4)):

$$\mathcal{M}^* = \operatorname{argmax}_{\mathcal{M}} \left(\sum_{(i,j) \in \mathcal{M}} s(i, j) \right). \quad (2)$$

Construction of the optimal one-to-one mapping set \mathcal{M}^* naturally leads to identifying the projection vector $\mathbf{v}_j|(x_j, y_j)$ of the orientation vector \mathbf{v}_j on the xy -plane of the j th mouse. Under the practically reasonable assumption that a mouse can only change the angle up to 90 degrees in the direction the mouse are looking (*i.e.*, mice do not stand on their hands), the optimal antenna index i_j^* for j th mouse can be found as follows:

$$i_j^* = \operatorname{argmin}_{i \in \{1, 2, 3, 4\}} E_Z [\| \mathbf{v}_j|(x_j, y_j) - \mathbf{a}_i \|], \quad (3)$$

where \mathbf{a}_i , $i = 1, 2, 3, 4$, is the antenna vector of the i th coil antenna (See supplementary Fig. 5). Finally, the optimal coil antenna that maximize the instantaneous system power transmission efficiency is determined by the principle of majority vote.

Reviewers' Comments:

Reviewer #3:

Remarks to the Author:

I thank the authors for taking my suggestions seriously. I feel it is much easier to understand what the authors did and how they achieved it.

Reviewers' Comments:

Reviewer #2:

Remarks to the Author:

The authors did a good job revisiting the data and conducted additional experiments to make sure the results are correct. I think the manuscript is now acceptable for publication. One minor comment is that the sentence "After 5 days of continuous PDT treatment, a 76 % decrease (combined), an 86 % increase (red), a 22 % decrease (purple), and 303 % increase (without LED) in tumor volumes (Fig. 5g)." is not a complete sentence. For example, they could add "... is observed."

Reviewer #2 (Remarks to the Author):

Comment: The authors did a good job revisiting the data and conducted additional experiments to make sure the results are correct. I think the manuscript is now acceptable for publication. One minor comment is that the sentence "After 5 days of continuous PDT treatment, a 76 % decrease (combined), an 86 % increase (red), a 22 % decrease (purple), and 303 % increase (without LED) in tumor volumes (Fig. 5g)." is not a complete sentence. For example, they could add "... is observed."

Response: We agreed with the reviewer's opinion. We modified the following sentence: "After 5 days of continuous PDT treatment, a 76 % decrease (combined), an 86 % increase (red), a 22 % decrease (purple), and 303 % increase (without LED) in tumor volumes are discovered (Fig. 5g)."